# Aberrant FGFR signaling mediates resistance to CDK4/6 inhibitors in ER+ breast cancer

Luigi Formisano[1], Yao Lu[1], Alberto Servetto[2], Ariella B. Hanker [1,2,3], Valerie M. Jansen[1], Joshua A. Bauer[4], Dhivya R. Sudhan[1,2], Angel L. Guerrero-Zotano [1], Sarah Croessmann[1], Yan Guo[5], Paula Gonzalez Ericsson [3], Kyung-min Lee[1], Mellissa J. Nixon[1], Luis J. Schwarz[1], Melinda E. Sanders[3,6], Teresa C. Dugger[1], Marcelo Rocha Cruz[7], Amir Behdad[7], Massimo Cristofanilli[7], Aditya Bardia[8], Joyce O'Shaughnessy[9], Rebecca J. Nagy[10], Richard B. Lanman [10], Nadia Solovieff[11], Wei He[11], Michelle Miller[12], Fei Su[12], Yu Shyr[5], Ingrid A. Mayer[1,3], Justin M. Balko [1] & Carlos L. Arteaga[1,2,3]

Using an ORF kinome screen in MCF-7 cells treated with the CDK4/6 inhibitor ribociclib plus fulvestrant, we identified FGFR1 as a mechanism of drug resistance. FGFR1-amplified/ER+ breast cancer cells and MCF-7 cells transduced with FGFR1 were resistant to fulvestrant ± ribociclib or palbociclib. This resistance was abrogated by treatment with the FGFR tyrosine kinase inhibitor (TKI) lucitanib. Addition of the FGFR TKI erdafitinib to palbociclib/fulvestrant induced complete responses of FGFR1-amplified/ER+ patient-derived-xenografts. Next generation sequencing of circulating tumor DNA (ctDNA) in 34 patients after progression on CDK4/6 inhibitors identified FGFR1/2 amplification or activating mutations in 14/34 (41%) post-progression specimens. Finally, ctDNA from patients enrolled in MONALEESA-2, the registration trial of ribociclib, showed that patients with FGFR1 amplification exhibited a shorter progression-free survival compared to patients with wild type FGFR1. Thus, we propose breast cancers with FGFR pathway alterations should be considered for trials using combinations of ER, CDK4/6 and FGFR antagonists.

[1] Departments of Medicine, Vanderbilt University Medical Center, Nashville 37232-6307 TN, USA. [2] UTSW Simmons Cancer Center, Dallas, TX 75230, USA. [3] Breast Cancer Program, Vanderbilt-Ingram Cancer Center, Vanderbilt University Medical Center, Nashville 37232-6307 TN, USA. [4] Departments of Biochemistry, Vanderbilt University Medical Center, Nashville 37232-6307 TN, USA. [5] Vanderbilt Center for Quantitative Sciences, Vanderbilt University School of Medicine, Nashville 37232-6307 TN, USA. [6] Departments of Pathology, Microbiology & Immunology, Vanderbilt University Medical Center, Nashville 37232-6307 TN, USA. [7] Robert H Lurie Comprehensive Cancer Center, Chicago 60611 IL, USA. [8] Massachusetts General Hospital Cancer Center, Harvard Medical School, Boston 02114 MA, USA. [9] Baylor University Medical Center, Texas Oncology, , US Oncology, Dallas 75246 TX, USA. [10] Guardant Health, Redwood City 94063 CA, USA. [11] Novartis Institutes for Biomedical Research, Cambridge 02139 MA, USA. [12] Novartis Pharmaceuticals Corporation, East Hanover 07936 NJ, USA. These authors contributed equally: Luigi Formisano, Yao Lu. Correspondence and requests for materials should be addressed to C.L.A. (email: carlos.arteaga@utsouthwestern.edu)

Despite significant treatment advances, breast cancer remains the second leading cause of cancer-related death in women. Approximately 70% of breast cancers express the estrogen receptor (ER)[1]. Antiestrogen (endocrine) therapies, such as selective estrogen receptor modulators (SERMs; i.e., tamoxifen), selective ER downregulators (SERDs; i.e., fulvestrant), and aromatase inhibitors are approved for the treatment of ER+ breast cancer[2]. To date, the only mechanisms of resistance to endocrine therapy that have been shown in the clinic are *ERBB2* (HER2) gene amplification[3,4] and, more recently, mutations in the ligand-binding domain (LBD) of ERα[5]. Recently, the addition of the CDK4/6 inhibitors palbociclib, ribociclib, and abemaciclib to aromatase inhibitors or to the ER downregulator fulvestrant have resulted in a markedly improved progression-free survival compared to the antiestrogen alone in patients with advanced ER+ breast cancer[6–10], leading to their approval by the FDA. Even though the great majority of patients with advanced disease treated with CDK4/6 inhibitors and antiestrogens benefit from this combination, virtually all eventually progress, underscoring the need to discover mechanisms of de novo and acquired resistance to this new standard of care.

To discover mechanisms of acquired resistance to antiestrogens plus CDK4/6 inhibitors, we expressed a library of 559 sequence-validated kinase open reading frame (ORF) clones[11] in ER+ MCF-7 cells treated with fulvestrant ± ribociclib and found that FGFR1 overexpression induces less sensitivity to this combination. We hypothesized that aberrant FGFR1 signaling is causally associated with a reduction of sensitivity to CDK4/6 inhibitors alone and in combination with antiestrogen therapy. In support of this hypothesis, we show herein that FGFR1 overexpression or amplification reduced the sensitivity to endocrine therapy ± palbociclib or ribociclib. Addition of FGFR TKIs or CCND1 siRNA to the combination increased drug sensitivity. Triple pharmacological inhibition of FGFR1, ERα and CDK4/6 induced complete

responses of patient-derived *FGFR1*-amplified ER+ xenografts. Further, FGFR1/2 alterations in plasma tumor DNA were detected at the time of progression on palbociclib in a small cohort of patients with advanced ER+ breast cancer. Finally, presence of FGFR1 alterations in baseline plasma tumor DNA was associated with a shorter progression free survival in the large randomized MONALEESA-2 trial of letrozole ± ribociclib, suggesting aberrant FGFR signaling is a potential mechanism of escape from endocrine therapy plus CDK4/6 inhibitors, a current standard of care in advanced ER+ breast cancer.

## Results

**ORF kinome screen identifies drivers of drugs resistance.** We expressed 559 sequence-validated kinase ORF clones, representing more than 70% of annotated kinases, in ER+ MCF-7 cells. ORF-expressing cells were treated with 10 nM fulvestrant (Fig. 1a) or 3 nM fulvestrant plus 250 nM ribociclib (Fig. 1b) and nuclei were scored to assess inhibition of proliferation; each treatment was performed in duplicate (see correlation in Supplementary Figure 1A). MCF-7 cells stably expressing a constitutively active MEK1 mutant (S218/222D or MEK-DD)[12] were used as drug-resistant positive controls (Supplementary Figure 1A, B). Fifteen ORFs from the fulvestrant screen and 17 ORFs from the fulvestrant plus ribociclib screen produced a robust *Z*-score ≥6 and were considered candidate resistance genes (Fig. 1c). To validate the 17 ORFs identified in the fulvestrant plus ribociclib screen, we transfected each of these kinases into MCF-7 cells and treated them with fulvestrant plus ribociclib and fulvestrant plus palbociclib over a range of drug concentrations. Protein overexpression as a result of transfection was confirmed by immunoblot analysis (Supplementary Figure 1B). Overexpression of 5/17 genes (FGFR1, CRKL, FGR, HCK, and FRK) conferred less sensitivity to both combinations at low (1 nM) and

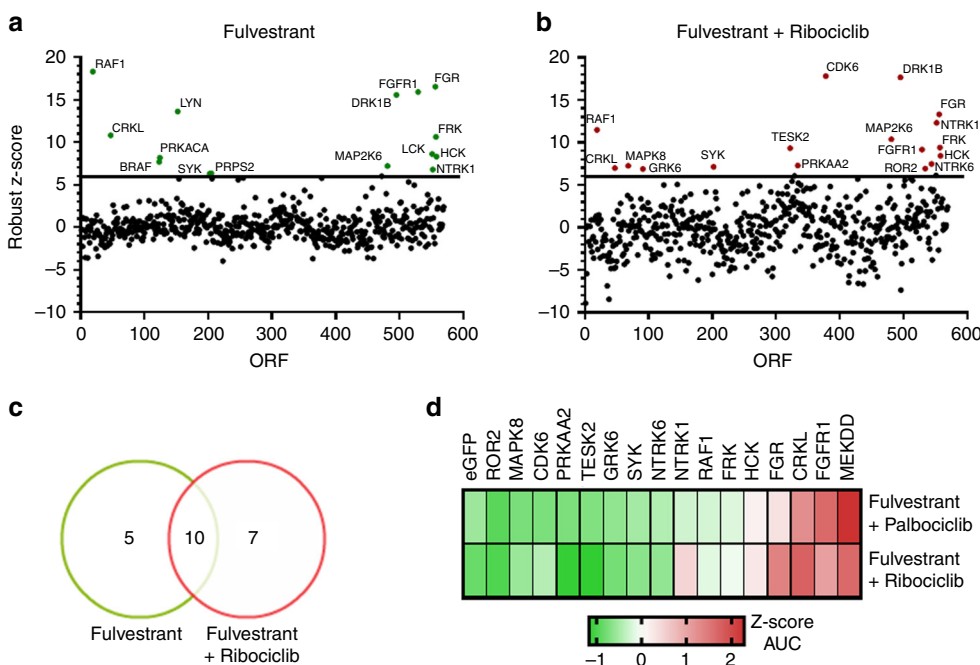

**Fig. 1** ORF kinome screen identifies drivers of resistance to fulvestrant ± CDK4/6 inhibitors. **a, b** ORF kinome screen in MCF-7 cells treated with fulvestrant (**a**) or fulvestrant plus ribociclib (**b**). ORFs are visualized by plotting the function $y = z$-score, $x =$ gene name. Data points represent the average of two replicate plates. **c** Modified Venn diagram of screening results showing the number of fulvestrant hits (left) or fulvestrant plus ribociclib hits (right). **d** Heat map displaying validation results of the hits selected in the fulvestrant plus ribociclib screen. Top and bottom row show the normalized area under the curve (AUC) for each candidate genes with 5-point drug concentration, fulvestrant plus palbociclib (top row) and fulvestrant plus ribociclib (bottom row). GFP and mutant MEK[S218/222D] (MEK-DD) were used as internal controls

high drug concentrations (1 μM) (Fig. 1d, Supplementary Figure 1C, Supplementary Table 1).

**FGFR1 overexpression promotes resistance to CDK4/6 inhibitors.** We next interrogated The Cancer Genome Atlas (TCGA, Cell 2015)[13] to determine the frequency of amplification and/or mRNA overexpression of the five validated hits in our screen. Among ER+ breast cancers, 15% harbored *FGFR1* gene amplification and/or mRNA overexpression whereas 2.5, 1.9, 2.4, and 1.7% harbored amplification and/or mRNA overexpression of *CRKL*, *HCK*, *FRK*, and *FGR*, respectively (Fig. 2a). *FGFR1* amplification is associated with early relapse and poor survival following adjuvant tamoxifen therapy and may be a potential resistance mechanism to endocrine therapy in ER+ breast cancer[14,15]. Consistent with these data, MCF-7 and T47D ER+ cells stably transduced with an FGFR1 expression vector were relatively resistant to fulvestrant plus palbociclib; this effect was completely reversed upon addition of the FGFR TKI lucitanib[16] (Fig. 2b, c, e). Of note, lucitanib alone did not significantly affect MCF-7[FGFR1] and T47D[FGFR1] cell growth. Fulvestrant and palbociclib were still able to downregulate ERα and p-RB levels, respectively, but only the triple combination of fulvestrant plus palbociclib plus lucitanib simultaneously reduced the levels of p-ERK1/2, ERα, and p-RB (Fig. 2d). These observations suggest that the overexpression of FGFR1 promotes resistance to fulvestrant ± palbociclib in ER+ breast cancer cells.

To determine if the FGFR inhibitor could overcome FGFR1-mediated effects in vivo, we implanted MCF-7[eGFP] and MCF-7[FGFR1] cells in ovariectomized athymic mice supplemented with a 14-day release 17β-estradiol pellet to support initial tumor establishment. Once tumors reached a volume of ≥200 mm³ and the estrogen pellet had expired (a state mimicking estrogen deprivation in patients treated with aromatase inhibitors), mice were randomized to treatment with vehicle, fulvestrant, or fulvestrant plus palbociclib ± lucitanib. In mice bearing MCF-7[FGFR1] xenografts, the triple combination was significantly more effective than the other treatments. In contrast, there was no statistical difference between the groups treated with fulvestrant plus palbociclib and the triple combination in mice with MCF-7[eGFP] xenografts (Fig. 3a). We next assessed pharmacodynamic biomarkers of drug action in tumor sections by immunohistochemistry (IHC). MCF-7[FGFR1] tumors showed higher levels of p-RB (Fig. 3b) and p-FGFR1 (Fig. 3c) relative to MCF-7[eGFP] tumors before treatment. In FGFR1-overexpressing tumors, only the triple combination markedly reduced p-RB and p-FGFR1 levels (fulvestrant plus palbociclib plus lucitanib vs. fulvestrant plus palbociclib; $p < 0.05$ and $p < 0.05$, respectively; Fig. 3b, c). On the other hand, the addition of lucitanib did not affect p-RB in MCF-7[eGFP] tumors (Fig. 3b, c). Lucitanib alone modestly delayed growth of MCF-7[FGFR1] xenografts, and the combination of fulvestrant plus lucitanib was less effective than fulvestrant plus palbociclib plus lucitanib ($p < 0.01$; Supplementary Figure 2A). Treatment with fulvestrant reduced ER levels in all arms (Supplementary Figure 2B). None of the treatments induced weight loss (Supplementary Figure 2C) and no signs of toxicity were observed.

**FGFR inhibition restores sensitivity to fulvestrant/palbociclib.** To determine whether naturally occurring *FGFR1* amplification promotes resistance to fulvestrant plus CDK4/6 inhibitors, we selected three ER+/HER2− human breast cancer cell lines with *FGFR1* gene amplification as determined by FISH: CAMA1, MDA-MB-134, and HC1500[15]. All three *FGFR1*-amplified cells were less sensitive to fulvestrant plus palbociclib or fulvestrant plus ribociclib than MCF-7 cells, where *FGFR1* is not amplified

(Supplementary Figure 3A). Indeed, in CAMA1 and MDA-MB-134 cells, but not in MCF-7 cells, the triple combination of fulvestrant/palbociclib/lucitanib was clearly superior at inducing growth arrest compared to fulvestrant alone or fulvestrant/palbociclib (Fig. 4a, b and Supplementary Figure 3B, C). Similar to lucitanib, knockdown of FGFR1 with two independent siRNAs also sensitized CAMA1 cells to fulvestrant/palbociclib (Supplementary Figure 4). Immunoblot analysis showed that only the triple combination simultaneously decreased p-RB, p-FRS2, p-ERK1/2, and ERα levels (Fig. 4c and Supplementary Figure 4D). Finally, the enhanced efficacy of the triple combination was confirmed using abemaciclib, a third and structurally distinct CDK4/6 inhibitor (Supplementary Figure 5).

Cell cycle analysis using flow cytometry showed that the addition of lucitanib to the combination of fulvestrant plus palbociclib significantly increased the percentage of cells in G0/G1 phase compared to fulvestrant plus palbociclib (93 vs. 85%; $p = 0.03$) and reduced the fraction of cells in S/G2M (19.7% vs. 12.6%; $p = 0.02$) (Fig. 4d). Of note, lucitanib alone or in combination with fulvestrant did not affect the cell cycle profile. We also observed a statistically significant increase in senescence-associated (SA) β-galactosidase-positive cells and H-Ras suppression upon treatment with the triple combination relative to cells treated with fulvestrant plus palbociclib (Fig. 4e–g), suggesting that the triple combination promotes both cell cycle arrest and cellular senescence.

To interrogate gene expression in cells where ER, CKD4/6, and FGFR1 are simultaneously inhibited, we performed RNA-seq of FGF2-stimulated CAMA1 cells treated with fulvestrant plus palbociclib ± lucitanib. Fulvestrant plus palbociclib downregulated the expression of 875 genes whereas treatment with the triple combination downregulated the expression of 3390 genes (Fig. 4h). We next analyzed differential signature enrichment using a set of 125 previously published breast cancer-related gene expression signatures[17,18]. Thirty different gene signatures were enriched in cells treated with fulvestrant plus palbociclib compared to cells treated with the triple combination (FDR <0.05). Treatment with fulvestrant/palbociclib/lucitanib reduced gene signatures involved in proliferation (such as CCND1), and intracellular signaling and mesenchymal pathways (such as RAS/ERK, IGF1, MYC, TGFβ, WNT_EMT) (Fig. 4i). Gene set enrichment analysis (GSEA) also showed increased expression of CCND1, Estrogen Receptor Early Response, and E2F1 gene signatures in FGF2-treated CAMA1 cells; treatment with lucitanib plus fulvestrant plus palbociclib completely suppressed activation of these pathways (Fig. 4j). These results suggest amplified FGFR1 signaling can promote escape from the growth inhibition induced by fulvestrant/palbociclib by activating transcriptional programs that sustain cell cycle progression.

The RAS/RAF/MEK/ERK pathway is a major effector of FGFR1 signaling[19]. Thus, we examined if inhibition of MEK/ERK would phenocopy the effect of the FGFR inhibitor on cell growth. Both lucitanib and selumetinib inhibited p-ERK1/2 levels. However, the combination of lucitanib/fulvestrant/palbociclib was more effective at blocking CAMA1 cell proliferation than the combination with fulvestrant/palbociclib/selumetinib (Supplementary Figure 6A–C), suggesting that FGFR hyperactivity can confer resistance independently of MEK/ERK activation.

**Cyclin D1 may mediate FGFR1-induced drug resistance.** RNA-seq analysis highlighted the overexpression of *CCND1*, a critical gene for cell cycle progression, as a possible mechanism of resistance to fulvestrant plus palbociclib. Notably, one-third of *FGFR1*-amplified tumors also harbor amplification of *CCND1*[20,21]. This co-amplification has also been associated with

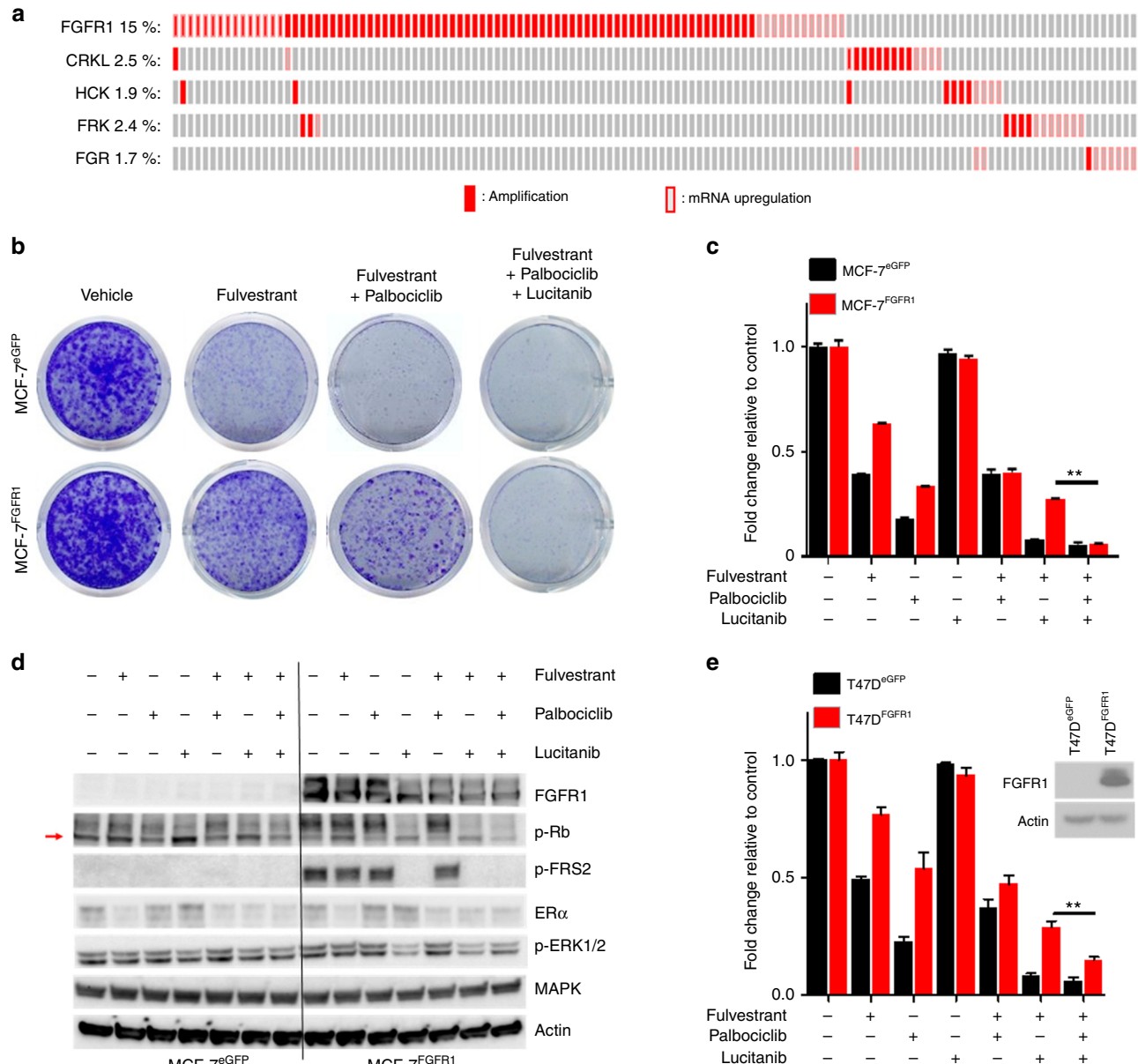

**Fig. 2** FGFR1 overexpression confers resistance to fulvestrant and palbociclib. **a** Tile plot of ER+ breast cancers in TCGA (Cell 2015) with amplification and/ or mRNA upregulation of *FGFR1*, *CRKL*, *HCK*, *FRK*, and *FGR*. **b**–**d** MCF-7eGFP and MCF-7FGFR1 cells were seeded in 6-well plates in full media supplemented with 2 ng/mL FGF2 and treated with vehicle (DMSO), 1 μM fulvestrant, or 1 μM fulvestrant plus 1 μM palbociclib ± 1 μM lucitanib. Drugs and media were replenished every 3 days. After 14 days, monolayers were stained with crystal violet and analyzed as described in Methods. Quantification of the integrated intensity values as fold change relative to vehicle-treated controls are shown (**c**) (**\*\***p < 0.01 vs. controls, Student's *t*-test). Cell lysates were subjected to immunoblot analyses with the indicated antibodies (**d**). **e** T47DeGFP and T47DFGFR1 cells were seeded in 6-well plates in full media supplemented with 2 ng/mL FGF2 and treated with vehicle (DMSO), 0.5 μM fulvestrant, or 0.5 μM fulvestrant plus 0.5 μM palbociclib ± 1 μM lucitanib. Drugs and media were replenished every 3 days. After 14 days, plates were washed and stained with crystal violet; imaging intensity was quantified by spectrophotometric detection. Quantification of the integrated intensity values as fold change relative to vehicle-treated controls are shown (**\*\***p < 0.01 vs. controls, Student's *t*-test). Lysates from T47D cells stably transduced with an FGFR1 expression vector were subjected to immunoblot analysis with FGFR1 and actin antibodies (on top-right of the panel)

resistance to estrogen deprivation in ER+ breast cancer and with poor patient outcome[20–22]. Of note, all three ER+/HER2−/FGFR1-amplified breast cancer cell lines used herein also harbor *CCND1* amplification. Of note, we were unable to identify any ER+/HER2−/FGFR1-amplified breast cancer cells without associated *CCND1* amplification. To investigate the role of CCND1 further, we next performed an 84-cell cycle gene PCR array. FGF2-stimulated CAMA1 cells showed upregulation of only six out of 84 genes in the array: *CCND1*, *CCND3*, *SERTAD1*,

*CDK6*, *GADD45A*, and *CDKN1A* (Fig. 5a). We interrogated the TGCA database to determine the expression of these six genes in ER+ tumors with *FGFR1* gene amplification and/or mRNA overexpression vs. ER+ tumors without FGFR1 alterations. Cyclin D1 (*CCND1*) was the only protein/gene significantly upregulated in ER+ tumors with *FGFR1* gene amplification and/ or mRNA overexpression (Supplementary Figure 7A, B). Higher levels of cyclin D1 mRNA and protein were also observed in *FGFR1*-amplified tumors without *CCND1* amplification

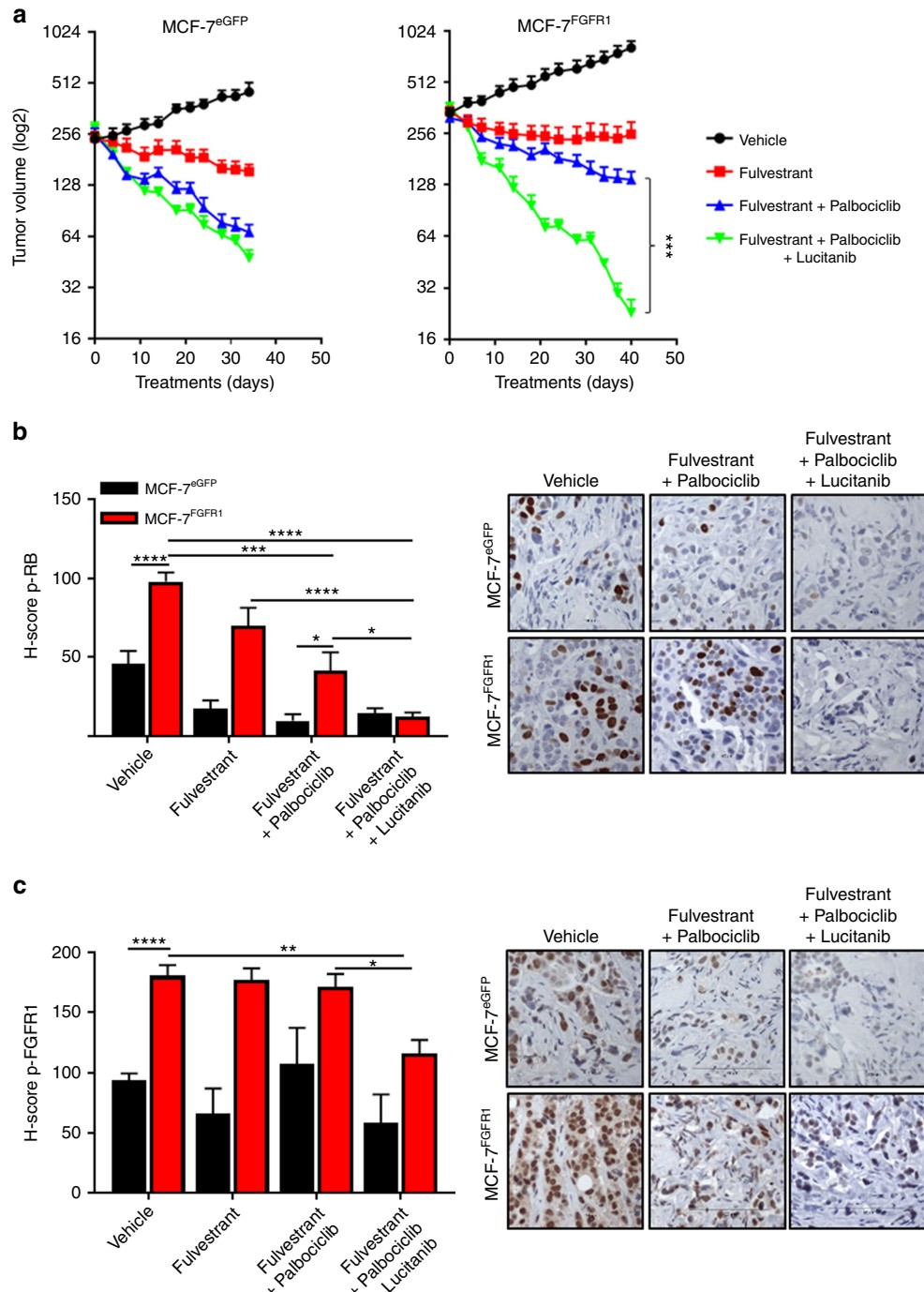

**Fig. 3** FGFR inhibitor lucitanib enhances the action of fulvestrant plus palbociclib against MCF-7[FGFR1] xenografts. **a** MCF-7[eGFP] (left) and MCF-7[FGFR1] (right) xenografts were established in ovariectomized athymic mice implanted with a s.c. 14-day release, 0.17-mg 17β-estradiol pellet. Once tumors reached ≥200 mm³, mice were randomized to treatment with vehicle, fulvestrant (5 mg/week), fulvestrant plus palbociclib (30 mg/kg/day p.o.), or fulvestrant plus palbociclib plus lucitanib (10 mg/kg/day p.o.). Each data point represents the mean tumor volume in mm³ ± SEM ($n = 8$ per arm, ****$p < 0.0001$ vs. single drug arms; Student's $t$-test). **b**, **c** MCF-7[eGFP] and MCF-7[FGFR1] tumors were harvested at the end of the treatment. Formalin-fixed, paraffin-embedded (FFPE) tumor sections were subjected to IHC with p-RB S807/811 (**b**) and p-FGFR1 Y653/4 (**c**) antibodies as described in Methods. The percent of p-RB+ and p-FGFR1+ tumor cells and their staining intensity were assessed by an expert breast pathologist (P.G.E.) blinded to treatment to generate an $H$-score. Total p-RB and p-FGFR1 $H$-scores are shown (*$p < 0.05$, **$p < 0.01$, ***$p < 0.001$, ****$p < 0.0001$; Student's $t$-test)

(Supplementary Figure 8C). Next, to further support a causal association between FGFR1 activation and cyclin D1 expression, we stimulated CAMA1 cells with FGF2. Addition of FGF2 induced cyclin D1 mRNA and protein levels as measured by RT-

PCR and immunoblot, respectively, and this induction was abrogated upon treatment with lucitanib (Fig. 5b, c).

We next examined whether high levels of cyclin D1 would negate the effect of FGFR inhibitors. Since CAMA1, HCC1500,

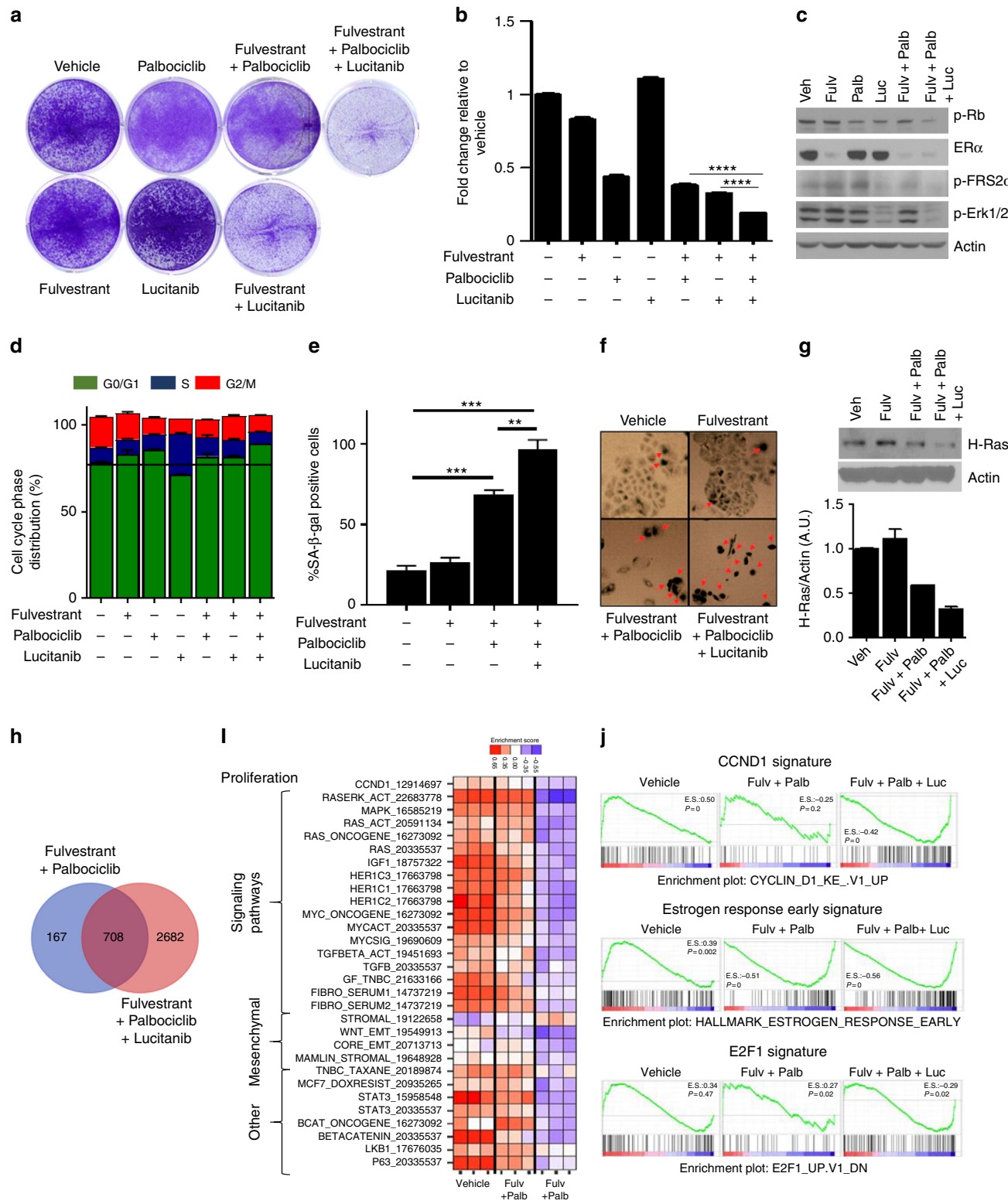

and MDA-MB-134 cells already exhibit *CCND1* amplification, we did not find rational to overexpress cyclin D1 in these cells. We first transduced MCF-7$^{FGFR1}$ and T47D$^{FGFR1}$ cells with a CCND1 expression vector, but this dual transfection approach was technically not feasible despite multiple attempts. Thus, we then tried a loss of function approach. Both knockdown of cyclin D1 with siRNA and treatment with lucitanib did not reduce CAMA1 cell growth in full media (not estrogen-depleted),

suggesting that the ER also plays a role in CAMA1 cell viability (Supplementary Figure 8A, B). In estrogen-depleted medium, however, transfection of cyclin D1 siRNA blocked the growth of CAMA1 cells to the same degree as FGFR1 knockdown (Supplementary Figure 8C). Immunoblot analysis showed that FGFR1 knockdown reduced cyclin D1 and pRB levels, similar to the knockdown of cyclin D1 (Supplementary Figure 10D). Furthermore, cyclin D1 siRNA in combination with fulvestrant

**Fig. 4** FGFR1 signaling sustains cell proliferation in *FGFR1*-amplified ER+ breast cancer cells treated with fulvestrant plus palbociclib. **a, b** CAMA1 cells were treated with vehicle (DMSO) or the indicated inhibitors (each at 1 μM) in FGF2-containing media. Cell media and inhibitors were replenished every 3 days. After 14 days, plates were washed and stained with crystal violet; imaging intensity was quantified by spectrophotometric detection. Representative images (**a**) and quantification of the integrated intensity values as fold change relative to vehicle-treated controls (**b**) are shown (****$p < 0.0001$ vs. controls, Student's *t*-test). **c** CAMA1 cells were treated as in **a** for 24 h. Cell lysates were prepared and subjected to immunoblot analysis with the indicated antibodies. **d** CAMA1 cells were serum-starved for 24 h and then treated with vehicle (DMSO) or each of the indicated inhibitors (all at 1 μM) in FGF2-containing media for 24 h. Following treatment, cells were stained with propidium iodide and analyzed by FACS. **e, f** CAMA1 cells were treated as above for 6 days. Representative images (**e**) and the percent of senescence-associated (SA)-β-galactosidase-positive cells per 5 high-power fields (**f**) are shown (**$p < 0.01$,***$p < 0.001$ vs. controls, Student's *t*-test). **g** CAMA1 cells were treated as in **e** for 6 days. Cell lysates were prepared and subjected to immunoblot analysis with the indicated antibodies. **h** Modified Venn diagram showing the number of downregulated genes ($q < 0.01$, log2 fold change > −0.5) in FGF2-stimulated CAMA1 cells treated with fulvestrant plus palbociclib (blue) or fulvestrant plus palbociclib plus lucitanib (FPL, red). **i** Heat map representing the modulation (FDR <0.05) of 30 gene expression signatures in FGF2-stimulated CAMA1 cells treated with fulvestrant plus palbociclib ± lucitanib for 6 h. **j** Enrichment plot for CCND1, Estrogen Response Early, and E2F1 signatures in CAMA1 cells

plus palbociclib was more effective at reducing cell growth than fulvestrant ± palbociclib or fulvestrant ± lucitanib in MCF-7[FGFR1] (Fig. 5d, e) and CAMA1 (Fig. 5f) cells, thus phenocopying the effect of FGFR inhibitors. Finally, MCF-7 cells stably transduced with a cyclin D1 vector (Fig. 5g) were completely resistant to fulvestrant ($p < 0.001$; Fig. 5h) and partially resistant to fulvestrant plus palbociclib ($p < 0.01$; Fig. 5i), with drugs tested over a dose range. Together, these data suggest at least a partial causal role of the FGFR1–cyclin D1 axis in resistance to antiestrogens alone and in combination with CDK4/6 inhibitors.

**Growth of FGFR1-amp ER+ patient-derived xenografts is blocked by triple combination**. Herein we examined the effect of the triple combination in mice bearing ER+/HER2−/*FGFR1*-amplified patient-derived xenografts (PDXs). Ovariectomized mice with established xenografts of >250 mm³ were treated with vehicle, fulvestrant, palbociclib, fulvestrant plus palbociclib, or fulvestrant plus palbociclib plus erdafitinib. Erdafitinib is a pan-FGFR inhibitor which, unlike lucitanib, does not inhibit VEGFR2[23]. In vitro, both erdafitinib and lucitanib showed similar effects against ER+/*FGFR1*-amplified CAMA1 cells (Supplementary Figure 9A, B). In vivo, only the triple combination was able to reduce tumor size by more than 50% after 3 weeks of treatment (Fig. 6a, b) with 3/10 xenografts achieving a complete response, which was sustained after treatment discontinuation. Ki67 levels were assessed by IHC (Fig. 6c, Supplementary Figure 10) and by flow cytometry (Supplementary Figure 11) after 1 week of treatment. By both methods, only the triple combination completely suppressed Ki67 levels (fulvestrant plus palbociclib vs. fulvestrant plus palbociclib plus erdafitinib; $p < 0.05$).

We next performed nanoString analysis by applying the nCounterPanCancer Pathways Panel to RNA from TM00386 PDXs harvested at the completion of therapy. This panel contains 770 genes from 13 cancer-associated canonical pathways including MAPK, STAT, PI3K, RAS, Cell Cycle, Apoptosis, Hedgehog, WNT, DNA Damage Control, Transcriptional Regulation, Chromatin Modification, and TGFβ. NanoString analysis showed that treatment with fulvestrant plus palbociclib reduced the expression of most cell cycle genes, but increased expression of *FGFR1*, *FGF12*, *FGF13*, and genes involved in MAPK signaling (Fig. 6d), suggesting they may represent an adaptive response to combined inhibition of CDK4/6 and ER. We were unable to perform nanoString analysis in xenografts treated with the triple combination because of the rapid tumor regression in response to the treatment. However, IHC of pharmacodynamic biomarkers showed that the triple combination reduced p-RB (Fig. 6e, Supplementary Figure 12A) and cyclin D1 levels much more potently than the other arms (Fig. 6f, Supplementary Figure 12B). Fulvestrant and erdafitinib downregulated ERα and p-FGFR1, respectively, (Supplementary Figure 12C, D), supporting optimal

dosing. Finally, immunoblot analysis of PDX lysates showed that the triple combination completely blocked ERK1/2 activation (Fig. 6g). TUNEL-positive cells were <1% in all treatment arms implying that, in the short term, apoptosis is not a major cellular mechanism of antitumor action of these drugs (Supplementary Figure 13A). No change in mouse weight was observed in any of the treatment arms (Supplementary Figure 13B). These results suggest that amplified FGFR1 signaling can promote tumor progression after CDK4/6 inhibitors and the addition of FGFR inhibitors can delay or abrogate this progression.

**FGFR pathway alterations are associated with poor outcome**. To examine the role of FGFR alterations on drug resistance in patients, we analyzed post-progression circulating tumor DNA (ctDNA) from 34 patients with ER+/HER2− breast cancer treated with palbociclib plus endocrine therapy. Plasma was analyzed for the presence of 114 mutations and copy number alterations and six gene fusions using the digital sequencing-based Guardant360 assay[24–26]. A significant number of the plasma samples harbored PIK3CA (15/34; 44%) and ESR1 (11/34; 32%) mutations. In 14/34 (41%) post-progression specimens, we detected FGFR alterations consisting on *FGFR1* amplification ($n = 9$), *FGFR2* amplification ($n = 2$), FGFR1 N546K ($n = 1$), FGFR2 N549K ($n = 1$), and FGFR2 V395D ($n = 1$) mutations. All three mutations have been reported as activating mutations[27–29]. Among these 14 plasma specimens, 2/14 exhibited CCNE1 amplification and 1/14 CDK6 amplification, alterations previously described as possible mechanisms of resistance to CDK4/6 inhibitors[30,31]. In five patients with *FGFR1* amplification ($n = 3$) and *FGFR2* amplification ($n = 2$), we were able to analyze pre-treatment ctDNA samples. For patients #1 and #4, we did not see a difference in the maximal mutant allele frequency (MAF), a surrogate of tumor shedding, between the pre- and post-treatment samples suggesting that for the patient #4, the amplification had been acquired during treatment (Supplementary Figure 14A). Moreover, six of nine tumors from patients with FGFR1 amplification in ctDNA were available for FISH (FGFR1: CEN8 ratio) or targeted capture next-generation sequencing (NGS). All of the six patients showed FGFR1 amplification. Similar to FGFR1, stable transduction of FGFR2 into MCF-7 cells reduced their sensitivity to fulvestrant plus palbociclib (Supplementary Figure 14B).

Finally, we investigated the correlation between detectable FGFR1 alterations in ctDNA and Progression Free Survival (PFS) in a cohort of patients enrolled in the MONALEESA-2 clinical trial[8]. This phase III trial tested the effectiveness of the aromatase inhibitor letrozole plus ribociclib vs. letrozole plus placebo in patients with ER+/HER2− advanced breast cancer. Median PFS was 16 months in the placebo group and 25.3 months in the ribociclib arm (hazard ratio (HR), 0.568; 95% confidence interval

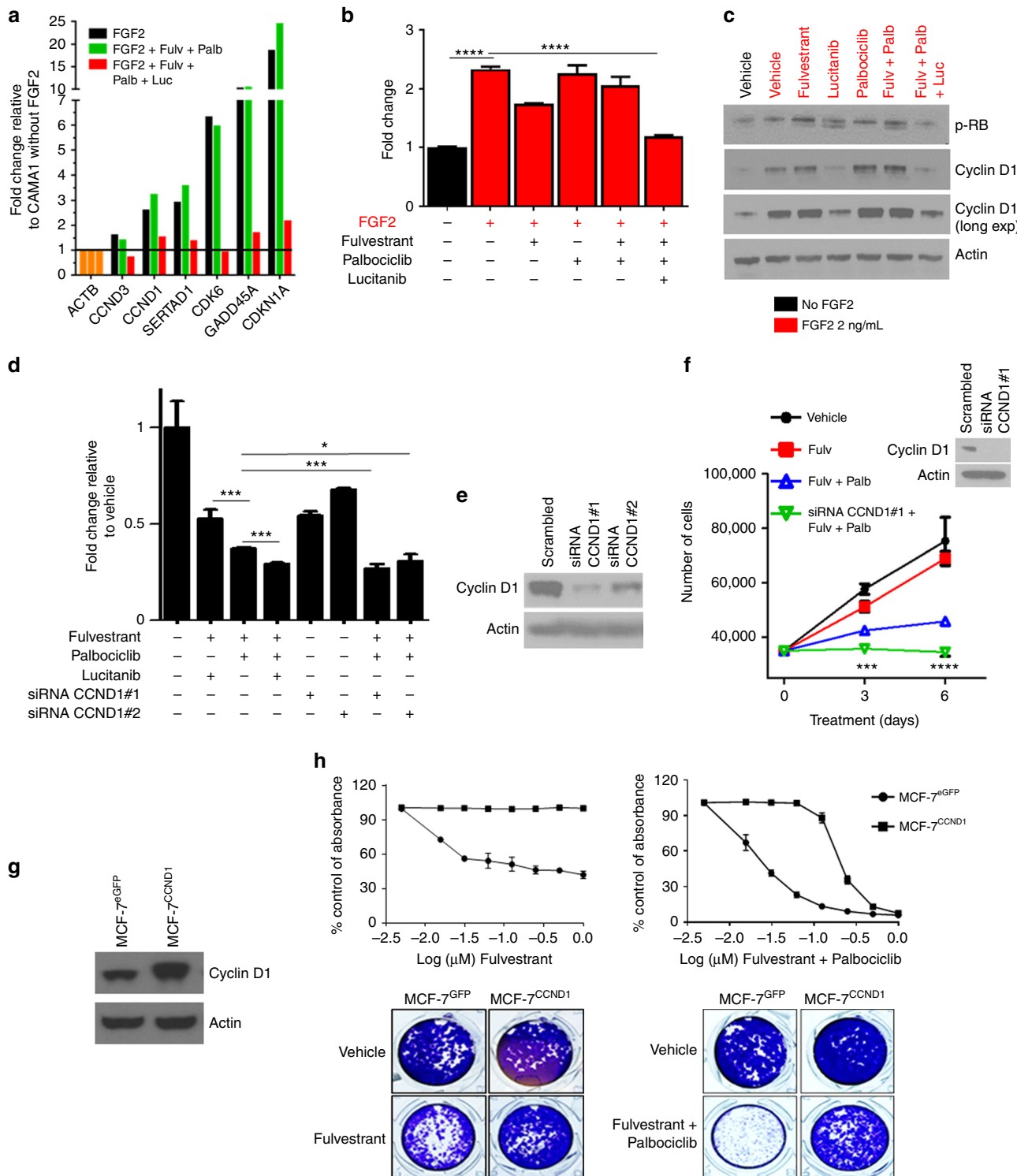

(CI): 0.457–0.704; $p = 9.63 \times 10^{-8}$)[32], resulting in the approval of ribociclib for use in patients with ER+ breast cancer. Region 8p11.23, where FGFR1/ZNF703 resides, was investigated in baseline ctDNA and found to be altered in 20 out of 427 patients (5%). A subgroup analysis of patients treated with ribociclib plus letrozole ($n = 212$) showed a PFS of 10.61 months in patients with FGFR1/ZNF703 amplification vs. 24.84 months in those with wild-type FGFR1/ZNF703 ($p = 0.075$; Fig. 7b). Although the patient sample size was small, this trend toward a lesser clinical benefit from ribociclib in patients with FGFR1/ZNF703 amplification warrants further evaluation in a larger sample size.

Baseline tumor samples from patients in MONALEESA-2 were assessed for mRNA expression using the NanoString 230-gene nCounter GX Human Cancer Reference panel. Evaluable mRNA expression data from baseline tumor samples were obtained from 391 of 668 (58.5%) randomized patients. To assess correlations between gene expression level and PFS, patients were classified into low and high mRNA expression subgroups using median expression (50%) as the cut-off. Patients with high FGFR1 mRNA expression that were treated with ribociclib plus letrozole showed a median PFS of 22.21 months. In patients with low FGFR1 mRNA, a median PFS has not been reached after a follow-up of

**Fig. 5** Cyclin D1 mediates FGFR1-driven resistance to fulvestrant plus palbociclib. **a** FGF2-stimulated CAMA1 cells were treated with the indicated inhibitors for 6 h. cDNA was analyzed using the RT$^2$ Cell Cycle PCR Array (Qiagen). **b, c** FGF2-stimulated CAMA1 cells were treated with the indicated inhibitors for 6 h. *CCND1* mRNA levels were analyzed by qRT-PCR (**b**) and cyclin D1 protein levels were analyzed by immunoblot (**c**). In **b**, each bar represents the mean *CCND1* transcript levels ± SD (****$p < 0.001$, Student's *t*-test). **d** MCF-7$^{FGFR1}$ cells in full media containing FGF2 were transfected with CCND1 or control siRNAs as described in Methods. Four days later, monolayers were harvested and cell counts determined using a Coulter Counter. Each bar represents the fold change relative to vehicle-treated controls (*$p < 0.05$, ***$p < 0.001$ vs. control siRNA, Student's *t*-test). **e** Cyclin D1 knockdown was confirmed by immunoblot analysis of cell lysates from plates treated as in **d**. **f** CAMA1 cells were transfected with CCND1 or control siRNAs as described in Methods. Full media containing FGF2 ± the indicated inhibitors was replenished every 3 days. Three and 6 days later, monolayers were harvested and cell counts determined using a Coulter Counter. Each data point represents the mean cell number ± SD of triplicate wells (***$p < 0.001$ vs. fulvestrant plus palbociclib, Student's *t*-test). Cyclin D1 knockdown was confirmed by immunoblot analysis of cell lysates from plates treated for 3 days (inset). **g** Lysates from MCF-7 cells stably transduced with a cyclin D1 expression vector were subjected to immunoblot analysis with cyclin D1 and actin antibodies. **h** MCF-7$^{eGFP}$ and MCF-7$^{CCND1}$ cells were plated in full media and treated with fulvestrant ± palbociclib over a dose range (0–1 μM) for 7 days. Cells were stained with crystal violet and monolayers quantified as described in Methods. Representative images at the concentration of 0.125 μM and quantification of the integrated intensity values as % of vehicle-treated controls are shown

32 months (HR 95%; CI: 0.56 (0.36–0.87); $p = 0.01$; Fig. 7b). This relationship was also observed for *CCNE1* mRNA high-expression (HR 95%; CI: 0.58 (0.37–0.89); $p = 0.01$) but not for CDK2 mRNA high-expression (HR 95%; CI: 0.83 (0.54–1.27); $p = 0.39$) (Supplementary Figure 15)[30]. Taken together, these data suggest a causal association between FGFR1 gene amplification and/or mRNA overexpression with early progression after treatment with CDK4/6 inhibitors and antiestrogens.

## Discussion

We report herein FGFR1 amplification/overexpression as a mechanism of resistance to treatment with CDK4/6 inhibitors in combination with antiestrogens. Using a screen with kinase ORFs, we identified five drug resistance candidates: FGFR1, CRKL, FGR, HCK, and FRK (Fig. 1). FGR, HCK, and FRK belong to the Src kinase family, which has been previously associated with antiestrogen resistance in breast cancer[33,34]. Overexpression and/or somatic alterations in these genes are rare (≤3%) in breast cancer (Fig. 2a). Further, the Src inhibitor dasatinib failed to increase the efficacy of endocrine therapy in ER+ breast cancer[35,36]. The screen also identified FGFR1 as a druggable candidate. *FGFR1* amplification and/or mRNA overexpression is present in around 15% of ER+ breast cancer (Fig. 2) and has been associated with early relapse following adjuvant tamoxifen and with poor survival in ER+ breast cancer[14]. In previous work before the adoption of CDK4/6 inhibitors, we reported that FGFR1 promotes estrogen-independent growth and resistance to endocrine therapy in ER+ breast tumors[15]. In this study, we extend these findings and describe a role for amplified *FGFR1* in resistance to the combination of CDK4/6 inhibitors plus endocrine therapy.

Previous studies have suggested acquired mutations of *RB1*, amplification of *CCNE*, amplification of *CDK6*, and/or suppression of the CDK2 inhibitors p27KIP1 and p21CIP1 as mechanisms of resistance to CDK4/6 inhibitors[30,31,37–39]. Recent works also suggests that inhibition of CDK4/6 is followed by activation of the PI3K/AKT/mTOR pathway as a mechanism of adaptive resistance[40–42]. Upon pharmacological inhibition of CDK4/6, upregulation of AKT signaling in breast cancer cells is linked to the accumulation of cyclin D1[30]. In turn, high cyclin D1 levels may contribute to resistance to CDK4/6 inhibitors by cyclin D1 associating with CDK2 and promoting progression into S phase of the cell cycle[30,43,44]. Indeed, the combination of PI3K and CDK4/6 inhibitors induces marked regressions of *PIK3CA* mutant xenografts[30,42,45], suggesting that blockade of PI3K and CDK4/6 is required to completely interrupt cyclin D1 function. RNA-seq data from CAMA1 cells showed that the triple combination of lucitanib/fulvestrant/palbociclib completely suppressed *CCND1*, Estrogen Receptor Early Response, and *E2F1*

gene signatures, suggesting that amplified FGFR1 signaling can promote escape from fulvestrant/palbociclib-induced growth inhibition by inducing cell cycle transcriptional programs. In addition, *CCND1* was one of the top six genes induced by FGF2 in CAMA1 cells (Fig. 5a). Further, *CCND1* is highly over-expressed in *FGFR1*-amplified breast cancers and the co-amplification of these two genes correlates with poor patient outcome[20]. Finally, knockdown of *CCND1* with siRNA markedly enhanced the effect of fulvestrant plus palbociclib against *FGFR1* amplified CAMA1 cells (Fig. 5f) and, similar to FGFR1, stable transduction of CCND1 into MCF-7 cells also induced resistance to fulvestrant plus palbociclib (Fig. 5g, h). These data suggest that cyclin D1 is a critical node in the resistance to CDK4/6 inhibitors mediated by aberrant FGFR signaling.

Upon treatment with CDK4/6 inhibitors, RB-competent cells have been shown to undergo quiescence or senescence[46,47]. Unlike quiescent cells, senescent cells do not return to the cell cycle following removal of the inhibitor and are considered refractory to other proliferation signals[48]. The transition from quiescence to senescence is distinguished by the downregulation of *MDM2*, redistribution of the chromatin-remodeling enzyme ATRX, and repression of *HRAS* transcription[46,49]. Our results showed that the addition of a FGFR TKI to fulvestrant/palbociclib statistically increased the number of SA β-galactosidase-positive cells and suppressed HRAS expression (Fig. 4e–g).

ctDNA analysis detected FGFR pathway alterations in 14/34 (41%) plasma specimens from patients at the time of progression on palbociclib. ctDNA identified three different FGFR activating mutations: FGFR1 N546K, FGFR2 N549K, and FGFR2 V395D. These data have important clinical implications as these gain of function mutations are responsive to small molecule FGFR inhibitors[50]. Within this cohort, 4/5 patients where pre- and post-treatment ctDNA was available, exhibited acquired *FGFR1/2* amplification. We recognize that use of ctDNA to detect copy number alterations is influenced by two factors: the number of copies in the tissue and the amount of DNA shedding into circulation. To reduce the number of biases, we normalized our results for aneuploidy within each patient sample. This normalization allows us to observe any variations in tumor shedding that may occur over the course of a patient's disease and therapy. Thus, the tumor shedding at baseline samples may not exhibit detectable amplification and that detection of amplification in subsequent samples may be due to an increase of shedding.

We also correlated FGFR1 amplification in ctDNA with clinical activity in 212 patients treated with ribociclib plus letrozole in the MONALEESA-2 study. Patients with FGFR1 amplification in ctDNA exhibited a trend toward a worse PFS when compared to patients without FGFR1 amplification (10.61 vs. 24.84 months, $p = 0.075$). TCGA reported FGFR1 amplification and/or

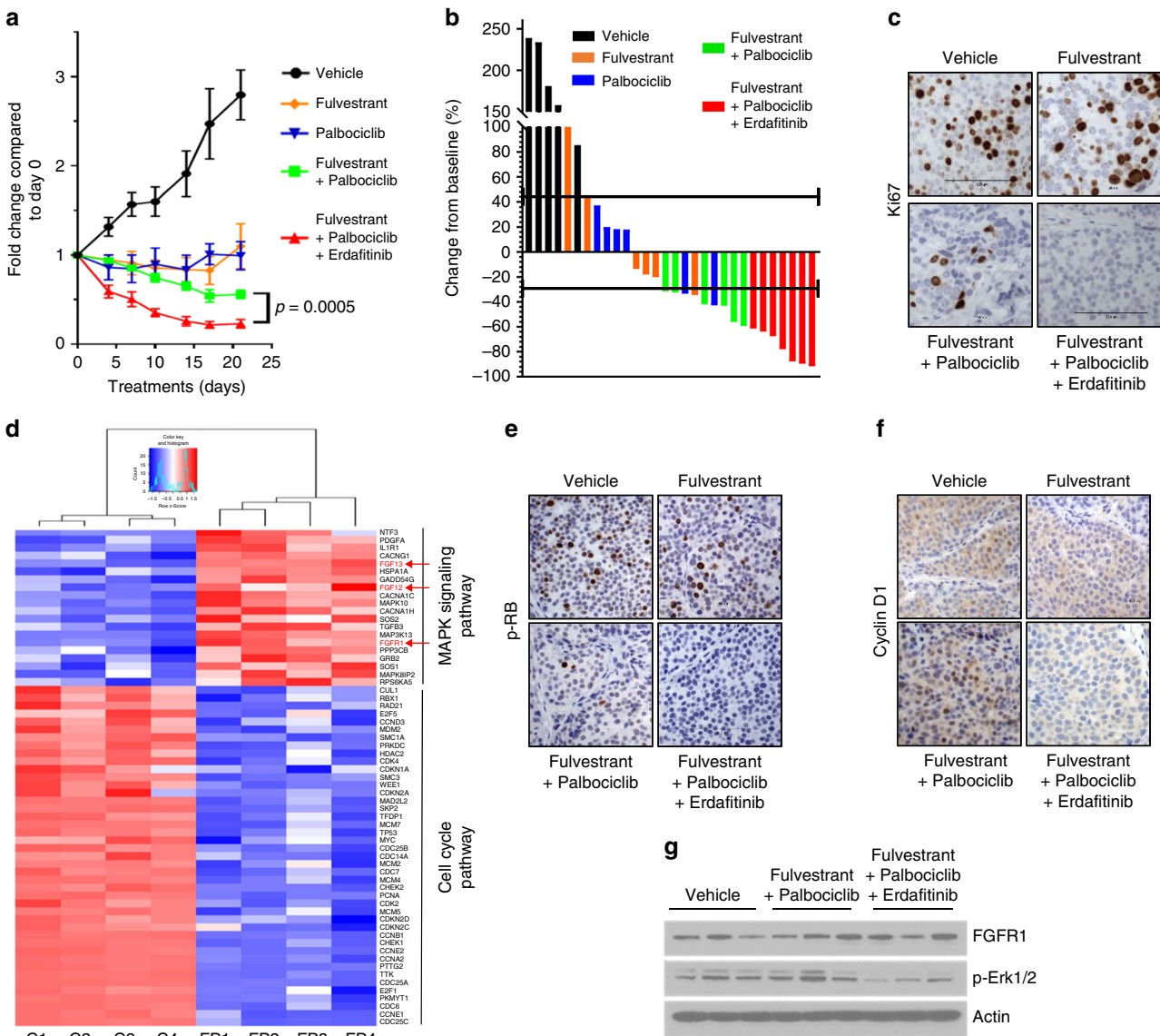

**Fig. 6** Combined inhibition of ER, CDK4/6, and FGFR1 inhibits the growth of ERα/*FGFR1*-amplified breast cancer PDXs. **a** ER+/HER2−/*FGFR1*-amplified TM00386 PDX fragments were established in ovariectomized SCID/beige mice supplemented with a 21-day release, 0.25-mg 17β-estradiol pellet. Once tumors reached ≥200 mm³, mice were randomized to the indicated treatment arms. Each data point represents the fold change in volume ± SEM (*n* = 8 per arm; ANOVA test). **b** Bar graph showing the % change in volume in individual xenografts after 3 weeks of treatment relative to individual tumor volumes on day 0. **c** Xenografts were harvested after 1 week of treatment and FFPE tumor sections were subjected to Ki67 IHC as described in Methods. The percent of Ki67+ tumor cells (inset) was assessed by an expert breast pathologist (P.G.E.) blinded to the treatment arm. **d** TM00386 PDXs in mice treated with vehicle or fulvestrant plus palbociclib were harvested and snap frozen at the end of treatment. RNA was extracted and subjected to nanoString analysis as described in Methods. Heat map represents different gene expression levels between controls (*n* = 4) and tumors treated with fulvestrant plus palbociclib (*n* = 4). **e**, **f** TM00386 PDXs were harvested at the end of treatment. FFPE sections from the PDXs were subjected to IHC with p-RB (**e**) and CCND1 (**f**) antibodies as described in Methods. The percent of p-RB and CCND1-positive tumor cells and their staining intensity were assessed by an expert breast pathologist (P.G.E.) blinded to the treatment arm to generate an *H*-score (shown in Supplementary figures). **g** TM00386 tumors were harvested at the end of treatment, 4 h after the last dose of palbociclib and erdafitinib and 24 h after the last dose of fulvestrant. Tumor lysates were prepared and subjected to immunoblot analyses with the indicated antibodies

overexpression in 15% of ER+ breast cancers whereas in MONALEESA-2, we only detected FGFR1 amplification in ctDNA in 5% of plasma specimens. This apparent discrepancy could be explained by several reasons. First, TCGA covered both *FGFR1* gene amplification and mRNA overexpression in primary tumors, whereas our analysis was limited to gene amplification in plasma tumor DNA using a less sensitive NGS assay. Second, TCGA data were generated from a broad cohort of primary ER+ breast cancers, whereas our cohort was from a more homogenous patient population with metastatic ER

+/HER2− breast cancer. It is possible, however, that detection of higher FGFR1 copies in ctDNA may be limited to those tumors with high FGFR1 gene amplification and that such group is the one predominantly associated with a refractoriness to CDK4/6 inhibitors. Finally, among 391 patients in MONALEESA-2 with tumor gene expression data, those with higher FGFR1 mRNA expression levels have so far exhibited a statistically significant shorter PFS than those with lower FGFR1 mRNA expression levels, consistent with and supportive of the ctDNA NGS finding.

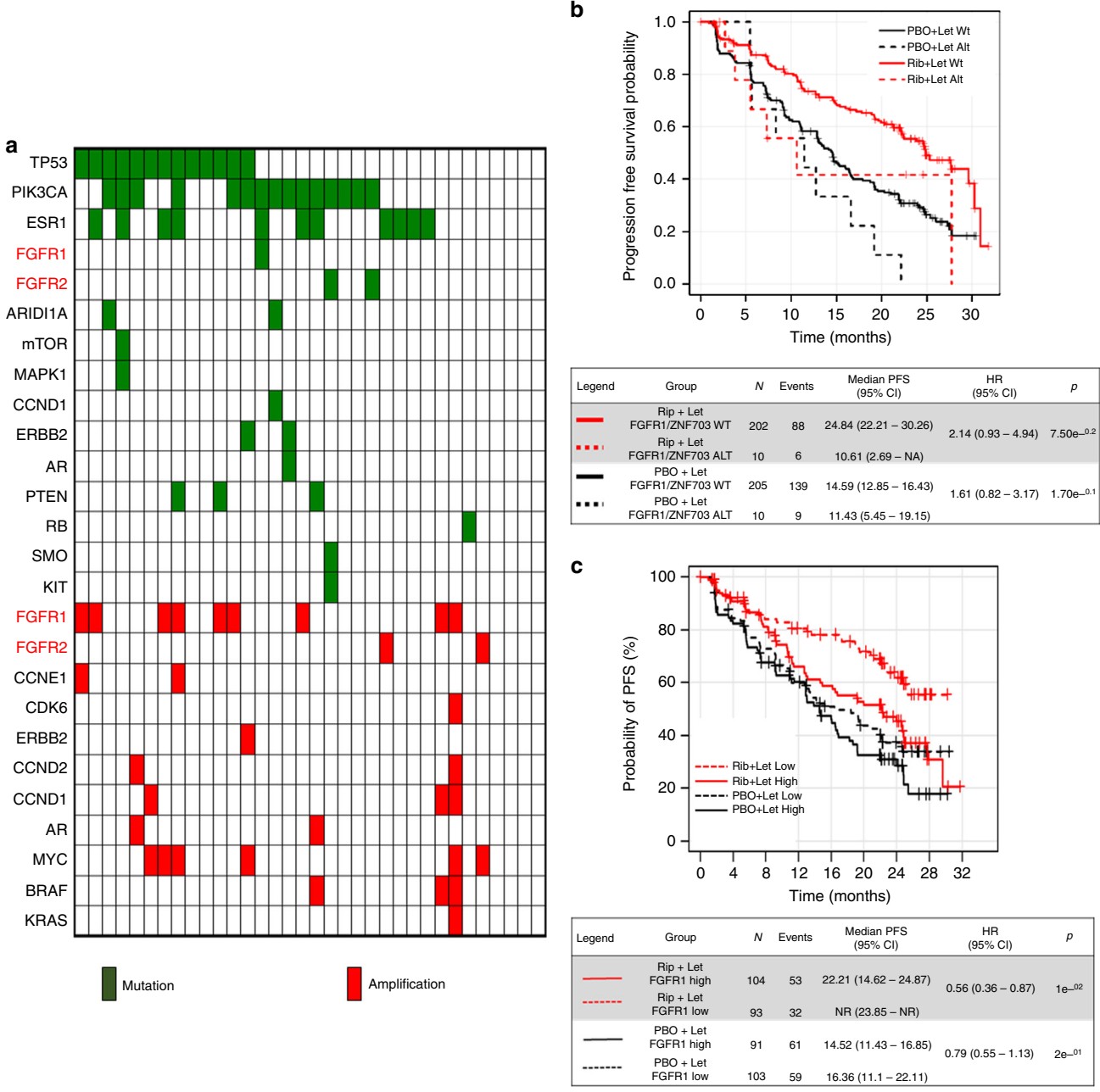

**Fig. 7** FGFR alterations correlate with poor outcome in ER+ breast cancers treated with CDK4/6 inhibitors and endocrine therapy. **a** Landscape of alterations in plasma tumor ctDNA from 34 patients progressing on palbociclib. **b** PFS in patients in the MONALEESA-2 trial of letrozole (Let) plus ribociclib (Rib) vs. letrozole plus placebo (PBO). Among patients treated in the investigational arm with letrozole plus ribociclib, those with detectable FGFR1/ZNF703 amplification (ALT) in ctDNA had a PFS of 10.61 months vs. 24.84 months in patients without FGFR1/ZNF703 amplification. **c** PFS in patients in MONALEESA-2 as a function of FGFR1 mRNA in archival tumor biopsies. Patients with cancers with high FGFR1 mRNA treated with letrozole/ribociclib exhibited a shorter PFS compared to patients with tumors with low FGFR1 mRNA

We recognize that resistance to CDK4/6 inhibitors is likely to be multifactorial. Indeed, patients progressing on palbociclib exhibited a relative high frequency of PIK3CA and ESR1 mutations in plasma tumor ctDNA (Fig. 7a). Nonetheless, these alterations were not found to be associated with a shorter PFS in the PALOMA-3 study of fulvestrant ± palbociclib[51]. Clearly, widespread molecular profiling of all tumors progressing on CDK4/6 inhibitors will be required to complete a landscape analysis of a potential plethora of mechanisms of acquired drug resistance, many of which will have implications for targeted treatment recommendations, including clinical trials, for these patients.

In summary, we describe herein a potential mechanism of resistance to CDK4/6 inhibitors in combination with endocrine therapy. Based on these results, we propose FGFR1 inhibitors in combination with ER and CDK4/6 antagonists as a testable therapeutic strategy in ER+ breast cancers also harboring aberrant FGFR signaling as a result of FGFR pathway gene alterations.

## Methods

**Cell lines and inhibitors**. MCF-7 (ATCC® HTB-22™), CAMA1 (ATCC® HTB-21™), HCC1500 (ATCC® CRC-2329™), and MDA-MD-134-VI (ATCC® HTB-23™) human breast cancer cells were obtained from the American Type Culture Collection (ATCC) in 2014 and maintained in ATCC-recommended media

supplemented with 10% FBS (Gibco) and 1× antibiotic/antimycotic (Gibco). Cell lines were authenticated by ATCC prior to purchase by the short tandem repeat (STR) method. All experiments were performed <2 months after thawing early passage cells. Mycoplasma testing was conducted for each cell line before use. Fulvestrant was provided by AstraZeneca Pharmaceuticals; ribociclib by Novartis; and erdafitinib by Janssen Pharmaceuticals. Lucitanib, palbociclib, and abemaciclib were purchased from SelleckChem.

**ORF lentiviral expression screen**. MCF-7 cells were plated in 384-well microplates (400 cells/well) in 10% DMEM-FBS. Cells were spin-infected the next day with the lentivirally packaged kinase ORF library in the presence of 8 μg/mL polybrene (Sigma). At 48 h post-infection, media were replaced with standard growth media or media containing either 10 nM fulvestrant, 3 nM fulvestrant, and 250 nM ribociclib, or 0.4 mg/mL puromycin. After 6 days, DAPI-stained cells were counted by high-content image counting using ImageXpress Micro Confocal System. The entire experiment was performed twice. A Z-score was calculated based on the following formula: $Z = \frac{X - \bar{x}}{S}$, where $X$ = test score; $\bar{x}$ = samples mean; $S$ = samples standard deviation.

**Viral transduction**. *CDK6, FGR, GRK6, RAF1, NTRK1, FRK, FGFR1, HCK, TESK2, NTRK3, PRKAA2, MAPK8, SYK, ROR2, CRKL*, and *GFP*-expressing lentiviral constructs were generated in the pLX302 Gateway vector (Open Biosystems). To generate stably transduced lines, 4 μg of pLX302-GFP (control) or kinase-encoding plasmid were co-transfected with 3 μg psPAX2 (plasmid encoding gag, pol, rev, and Tat genes) and 1 μg pMD2G envelope plasmid (Sigma Aldrich) into 293FT cells using Lipofectamine 2000 (Thermo Fisher). 293FT medium was changed 24 h post-transfection. Virus-containing supernatants were harvested 48 and 72 h post-transfection, passed through a 0.45-μm filter, diluted 1:4, and applied to target cells with 8 μg/mL polybrene (Sigma Aldrich). Transduced cells were selected in 1 μg/mL puromycin.

**Clonogenic assays**. Cells ($5 \times 10^4$/well) were seeded in triplicate in 10% DMEM-FBS (full media) in 6-well plates and treated with ±2 ng/mL FGF2 (Sigma) and ±1 μM of indicates drugs. Media, FGF2, and drugs were replenished every 3 days until control wells reached 50–70% confluency. Monolayers were then fixed and stained with 20% methanol/80% water/0.5% crystal violet for 20 min, washed with water, and dried. After photographic images of the plates were obtained, the crystal violet was solubilized with 20% acid acetic and the image intensity of the monolayers was quantified by spectrophotometric detection at 490 nm using a plate reader (Glo-Max®-Multi Detection System, Promega).

**siRNA transfections**. Cells were reverse transfected using Lipofectamine RNAi-MAX® (Invitrogen) and 25 nM siRNA (siControl—Ambion cat. #4390843; si*FGFR1*#1—Ambion cat. #AM16708; si*FGFR1*#2—Ambion cat. #AM16704; si*CCND1*#1—Thermo-Fisher: sense sequence: CCAGAGUGAUCAAGUGUGAUU, antisense sequence: UCACACUUGAUCACUCUGGUU; si*CCND1*#2—Thermo-Fisher cat. 4392420). The next day, cells were seeded in 10% DMEM-FBS or IMEM/10% dextran-charcoal-treated FBS (DCC-FBS; Hyclone, contains <0.0367 pM 17β-estradiol (E2)) + 2 ng/mL FGF2 ± inhibitors in either 6-well plates ($5 \times 10^4$/well for proliferation assays) or in 60-mm plates (for immunoblot analysis). Cell media ± FGF2 were changed every 3 days thereafter. Cells were trypsinized 6 and 9 days post-transfection and counted using a Coulter Counter (Beckman Coulter). For immunoblot analyses, the cells were harvested and protein lysates prepared on day 3 post-transfection.

**β-galactosidase staining**. Cells ($2 \times 10^5$/well) were plated in 6-well plates in triplicate in media containing 10% FBS + 2 ng/mL FGF2 and then treated with DMSO, or 1 μM fulvestrant or 1 μM fulvestrant plus 1 μM palbociclib ± 1 μM lucitanib. Media, FGF ligands, and drugs were replenished every 3 days. After 7 days, cells were stained with β-galactosidase (Cell Signaling Technology) at pH 6.0 following the manufacturer's protocol. Cells were photographed and β-galactosidase-positive cells were counted manually using a light-field microscope.

**Immunoblot analysis**. Cells were lysed with RIPA buffer (150 mM NaCl, 1.0% IGEPAL®, 0.5% sodium deoxycholate, 0.1% SDS, and 50 mM Tris, pH 8.0. [Sigma] and 1× protease inhibitor cocktail [Roche]). Whole cell lysates were separated by SDS-PAGE, transferred to nitrocellulose, and subjected to immunoblot analyses using primary antibodies against ERα(F-10) #sc-8002 1:200, H-RAS sc-29 1:200, cyclin D1 sc-450 1:200 (Santa Cruz Biotech.), phosphorylated RB #8516 1:1000, phosphorylated FRS-2α #3861 1:500, phosphorylated ERK1/2 #9101 1:1000, ERK1/2 #4695 1:1000, FGFR2 #23328 1:500, actin #4970 1:2000 (Cell Signaling), and FGFR1 #ab829 1:500 (Abcam). HRP-conjugated anti-rabbit and anti-mouse were used as secondary antibodies (Santa Cruz Biotechnology). Immunoreactive proteins were visualized by enhanced chemiluminescence (Pierce, Rockford, IL, USA). Membranes were cut horizontally to probe with multiple antibodies. Blots probed with phospho-antibodies were stripped with Restore Western Blot Stripping Buffer (Thermo Fisher Scientific) and re-probed with antibodies to the total protein.

Uncropped images of the most relevant immunoblots are shown in Supplementary Figure 16.

**Cell cycle PCR array**. CAMA1 cells were plated in full media ± 2 ng/mL FGF2 and then treated with 1 μM fulvestrant plus 1 μM palbociclib ± 1 μM lucitanib for 6 h. Cells were harvested and RNA was purified using a RNA purification kit (Maxwell, Promega). cDNA was generated using High Capacity cDNA Reverse Transcription Kits (Applied Biosystems, Carlsbad, CA) followed by analysis of 84 cell cycle pathway genes using the Cell Cycle PCR Array (Qiagen, PAHS-020Z). The original data are archived in Dryad Digital Repository (https://doi.org/10.5061/dryad.10tj37m).

**Flow cytometry**. Cells were incubated in serum-free media for 24 h and then treated with media containing 10% FBS + 2 ng/mL FGF2 ± inhibitors for 24 h. Next, cells were washed with PBS and fixed in 99% methanol for 3 h at −20 °C. Cells were then incubated with 0.1 mg/mL RNase A (Qiagen) and 40 μg/mL propidium iodide (PI; Sigma-Aldrich) for 10 min at room temperature. Fluorescence-activated cell sorting (FACS) analysis was performed on the LSRFortessa X-20 Cell Analyzer (BD Biosciences) and the data were analyzed with FlowJo software.

**Ki67 detection and cell sorting**. TM00386 PDXs established in ovariectomized SCID/beige mice were harvested and dissociated using digestion buffer [125 μg/mL DnaseI (#LS002138, Worthington), 10 μg/mL Dispase (#LS02109, Worthington), 500 μg/mL Collagenase 3 (#LS004182, Worthington), and 5× triple antibiotics (#15240-062, Invitrogen)] to generate single cells. The cells (~$2 \times 10^6$) were washed with PBS and fixed in 99% methanol for 3 h at −20 °C. Cells were then incubated with 20 μL of PE Mouse anti-Human Ki67 antibody (#51-36525, BD Pharmingen) or 20 μL of PE Mouse IgG1 (#51-35405, BD Pharmingen). FACS analysis was performed on the LSRFortessa X-20 Cell Analyzer (BD Biosciences), and the data were analyzed with FlowJo software. The remaining half of the PDXs was fixed in 10% neutral buffered formalin followed by embedding in paraffin for Ki67 IHC.

**NanoString analyses**. *TM00386 PDX*: Total RNA was extracted from TM00386 frozen tumors using Maxwell® (Promega). Total RNA was quantified using the Quant-iT Ribo-Green RNA Assay Kit (Invitrogen) and normalized to 4 ng/mL. Gene expression was measured using the nanoString nCounter PanCancer Profiling Panel (nanoString Technologies) following the manufacturer's instructions. Data were analyzed by nanoString nSolver Analysis Software v2.0.

*MONALEESA-2 trial*: Archival tumor samples were collected and assessed for mRNA expression using the NanoString 230-gene nCounter® GX Human Cancer Reference panel. To evaluate correlations between gene expression levels and PFS, patients were classified into low and high FGFR1 mRNA subgroups using median expression (50%) as the cut-off. Kaplan–Meier estimator was used to identify the survival probability of patients at each time point. Cox proportional hazards models were used to estimate HRs including 95% CIs of ribociclib vs. letrozole in the FGFR1 high and low expression subgroups. The original data regarding the nanostring analysis in TM00386 PDX are shown in Supplementary_Data_1, and the nanostring analysis in MONALEESA-2 trial are shown in Supplementary_Data_2.

**Xenograft studies**. Mouse experiments were approved by the Vanderbilt Institutional Animal Care and Use Committee and the experiment was performed according with all relevant ethical regulations. Female ovariectomized athymic mice (Harlan Sprague Dawley) were implanted with a 14-day-release 17β-estradiol pellet (0.17 mg; Innovative Research of America). The following day, $1 \times 10^7$ MCF-7$^{eGFP}$ or MCF-7$^{FGFR1}$ cells suspended in IMEM and Matrigel at 1:1 ratio were injected subcutaneously (s.c.) into the right flank of each mouse. Approximately 4 weeks later, mice bearing tumors measuring ≥200 mm³ were randomized to treatment with (1) vehicle (control), (2) fulvestrant (5 mg per week; s.c.), (3) fulvestrant plus palbociclib (30 mg/kg/day via orogastric gavage), or (4) fulvestrant plus palbociclib plus lucitanib (10 mg/kg/day via orogastric gavage). In a second experiment, we used an ER+/HER2−/FGFR1-amplified breast cancer PDX (TM00386; Jackson Laboratory) (http://tumor.informatics.jax.org/mtbwi/pdxDetails.do?modelID=TM00368). Tumor fragments were implanted s.c. in the right flank of female ovariectomized SCID/beige mice (Jackson Laboratory) implanted with a 21-day release, 0.25-mg 17β-estradiol pellet. Tumor fragments were serially transplanted in athymic or SCID/beige mice under general anesthesia. When xenografts reached an average size of ≥200 mm³, mice were randomized to treatment with (1) vehicle, (2) fulvestrant, (3) palbociclib, (4) fulvestrant plus palbociclib, or (5) fulvestrant plus palbociclib plus erdafitinib (12.5 mg/kg via orogastric gavage twice a day). Animal weights and tumor diameters (with calipers) were measured twice weekly and tumor volume in cubic millimeters was calculated with the formula: volume = width² × length/2. After 6 weeks, tumors were harvested and snap-frozen in liquid nitrogen or fixed in 10% neutral buffered formalin followed by embedding in paraffin for IHC. Tumors were harvested 4 h after the last dose of palbociclib or erdafitinib or lucitanib or 24 h after the last dose of fulvestrant. Five-micrometer paraffinized sections were used for IHC using p-FGFR1 Y653/54 (Abcam #111124), p-RB S807/811 (Cell Signaling #8516), and ERα (Santa Cruz Biotech #8002). Sections were scored by an expert breast pathologist (P.G.E.) blinded to the treatment arm.

**Targeted sequencing of ctDNA.** Circulating cell-free DNA (cfDNA) was obtained from whole blood of a cohort of patients with advanced ER+ breast cancer treated at three different USA centers with the standard of care approved CDK4/6 inhibitors where the providers requested a plasma tumor ctDNA NGS test from Guardant Health; 5–30 ng of cfDNA were isolated. Analysis of cell tumor (ct) DNA was performed using a commercially available digital NGS assay (Guardant360, Guardant Health, Inc., Redwood City, CA). Guardant360 is a 73-gene ctDNA NGS panel from a CLIA-licensed, CAP-accredited laboratory. It provides complete exon sequencing of 19 cancer genes, critical exons of 54 genes and amplifications (18 genes), fusions (six genes), and indels (23 genes) with high clinical sensitivity rates (85% in stage III/IV solid tumors) and ultra-high specificity (>99.9999%)[52]. The original data are shown in Supplementary_Data_3.

**NGS on ctDNA.** Blood samples were collected from patients in K2 EDTA tubes and processed into plasma using two centrifugation steps. ctDNA was extracted from 2 mL of patient plasma using the Qiagen Circulating Nucleic Acid kit (Qiagen, CA). Total extracted ctDNA was used to generate NGS libraries via end repair, A-tailing, and adapter ligation using the Illumina TruSeq Nano DNA Library Prep kit (Illumina, CA). Libraries were enriched for a specific 2.9 Mb of the human genome designed to contain ~600 genes relevant to cancer using Agilent SureSelect XT Custom baits and SureSelectXT capture enrichment reagents (Agilent, CA). Captured libraries were combined in equimolar pools and sequenced on the Illumina HiSeq 2500 sequencer, targeting ≥70 million sequencing reads per sample to ensure that unique coverage of the PanCancer panel exceeds 1000× or approaches the maximal complexity of the sequencing library. Sequence reads were aligned to the reference human genome (build hg19) using the Burrows–Wheeler Aligner (BWA-MEM)[53]. The Genome Analysis ToolKit was used for local realignment and base quality score recalibration[54,55]. Single nucleotide variants were identified with MuTect;[56] copy number variants were called with custom calling and normalization methods using depth of coverage; indels were called using PINDEL;[57] translocations were called using SOCRATES[58]. The original data are shown in Supplementary_Data_4.

**RNA-seq and cDNA library construction.** CAMA1 cells were plated in full media ± 2 ng/mL FGF2 (Sigma) and then treated with DMSO or 1 μM fulvestrant plus 1 μM palbociclib ± 1 μM lucitanib for 6 h. Cells were harvested and RNA was purified using a RNA purification kit (Maxwell, Promega). Total RNA was quantified using the Quant-iT Ribo-Green RNA Assay Kit (Invitrogen) and normalized to 4 ng/mL; 200 ng of each sample were used for library preparation in an automated variant of the Illumina Tru Seq RNA Sample Preparation protocol (Revision A, 2010). This method uses oligo(dT) beads to select mRNA from the total RNA sample and is followed by heat fragmentation and cDNA synthesis from the RNA template. The resultant cDNA went through library preparation (end repair, base "A" addition, adapter ligation, and enrichment) using Broad Institute–designed indexed adapters for multiplexing. After enrichment, libraries were quantitated with qPCR using the KAPA LibraryQuantification Kit for Illumina Sequencing Platforms and pooled equimolarly. The entire process was performed in a 96-well format with all pipetting done by either the Agilent Bravo or PerkinElmer JANUS Mini liquid handlers.

**Gene expression analyses.** We generated rlog-transformed count data using DESeq2, filtering low expressing genes (<10% samples with 0 count and mean >4). This resulted in 17,862 transcripts that served as input for the following analysis: (1) Single-sample gene set enrichment for 125 previously published breast cancer-related gene expression signatures calculated using the GSVA (gene set variation analysis) software. The original data are shown in Supplementary_Data_5.

**Statistical analyses.** Results are representative of three independent experiments and are expressed as the mean ± SEM. A $p$ value of <0.05, determined by Student's $t$-test or ANOVA test as indicated in the figure legends, was considered statistically significant.

## Data availability

The original data regarding the gene expression and sequencing data (RNA-seq, GE analysis, PCR array, Nanostring analysis, Targeted seq of ctDNA, NGS on ctDNA) are archived as indicated in each specific paragraph of the Methods section. The data that support the findings of this study are available from the corresponding author upon reasonable request.

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

## Acknowledgements

This study was supported by NIH Breast SPORE grant P50 CA098131, Vanderbilt-Ingram Cancer Center Support grant P30 CA68485, Susan G. Komen for the Cure Breast Cancer Foundation grant SAC100013, and a grant from the Breast Cancer Research Foundation. V.M.J. was supported by Conquer Cancer Foundation ASCO Young Investigator Award 8364, Susan G. Komen Postdoctoral Fellowship Grant PDF15329319, and the Vanderbilt Clinical Oncology Research Career Development Program (2K12CA090625-17). J.A.B. was supported by NCI Research Specialist Award R50 CA211206. L.F. was supported by AIRC under MFAG 2018 – ID. 21505.

## Author contributions

Experimental study design/conception: L.F., Y.L. and C.L.A. Data acquisition and analysis: L.F., Y.L., A.S., A.B.H., V.M.J., J.A.B., D.R.S., A.L.GZ., S.C., Y.G., P.G.E., K.L., M.J.N. and L.J.S. Writing of manuscript: L.F., F.S. and C.L.A. Clinical data analysis: M.E.S., T.C.D., M. R.C., A.Ba., M.C., A.Be., J.O., R.J.N., R.B.L., N.S., W.H., M.M., F.S., Y.S., I.A.M. and J.M.B. Review of manuscript: all authors.

## Additional information

**Competing interests:** C.L.A. receives grant support from Pfizer, Lilly, Bayer, and Radius. He serves in advisory roles to Symphogen, Daiichi Sankyo, TAIHO Oncology, Novartis, Merck, PUMA Biotechnology, Lilly, Radius, Sanofi, OrigiMed, and H3Biomedicine. He serves in the Scientific Advisory Board of the Komen Foundation. He holds stock options in Provista and Y-TRAP. R.J.N and R.B.L. are employees of Guardant Health; N.S., M.M., and F.S. are employees of Novartis Pharmaceuticals Corporation. The remaining authors declare no competing interests.

