## [Peer Review File · Nature Communications]

Reviewers' comments:

Reviewer #1 (Remarks to the Author):

Aberrant FGFR signaling mediates resistance to CDK4/6 inhibitors in ER+ breast cancer

Formisano L, Lu Y et al

In the following manuscript the authors expressed the ORFs of 559 kinases in MCF7 cells treated with fulvestrant \pm ribociclib and identified 11 kinases which increased cell viability by greater than 30%. Five kinases (FGFR1, FRK, HCK, FGR, CRKL) also induced resistance in secondary screens. Transduction of MCF7 and T47D with FGFR1 reduced response to ribociclib \pm fulvestrant, an observation also noted in FGFR1-amplified breast cancer cell lines. The authors show treatment of an FGFR1-amplified PDX with fulvestrant, palbociclib or both delayed tumour growth but the addition of erdafitinib as a triple combination resulted in complete tumour response. The authors provide correlative evidence that treatment of FGFR1-amplified cells with FGF2, enhances expression of CCND1 associated gene signature and that siRNA of CCND1 increased sensitivity of FGFR1-amplified cells to fulvestrant and palbociclib. The authors proceed to show using NGS of ctDNA from 34 patients pre and post progression on CDK4/6 inhibitors that activating mutations in FGFR1/2 were evident in 41% of the post-progression samples. They then show that ctDNA analysis of MONALEESA-2 showed patients with FGFR1 amplification show PFS of 10.61 months versus 24.84 months in those harbouring wt-FGFR. The authors conclude that ER+ breast cancers, which harbour FGFR1 amplification are resistant to fulvestrant and CDK4/6 inhibitors and may benefit from a triple combination targeting ER, CDK4/6 and FGFR1 signalling.

Major Comments

The manuscript is well written but is largely correlative and the authors themselves highlight many limitations within their study particularly with regard to the translational science using MONALEESA-2.

With regard to target identification using the ORF 559 kinase library it is of interest that FGR, HCK, FRK all members of the src family, which when over expressed showed decreased sensitivity to ribociclib and palbociclib yet they were omitted from further study by nature of the fact dasatinib has not shown utility in endocrine resistant breast cancer patients and therefore the authors selected FGFR1 as their preferred target which they support by using TCGA and the fact that FGFR1 amplification is already associated with resistance to tamoxifen.

The authors go on to test the antiproliferative effect of lucitanib an FGFR1 and VEGFR inhibitor. They show that a triple combination of CDK4/6 inhibition with fulvestrant shows greatest effect (Fig 2). However, the statement that FGFR1 inhibition alone is ineffective in the colony forming assays shown in Suppl 2 is a little misleading as it appears there is a least a 40% reduction in the MCF7FGFR1, yet the T47DFGFR1 is quantified and the WB confirming expression is also only shown for T47D. The data should be provided for both overexpressing cell lines. Nonetheless the triple does indeed appear more effective. The authors then state that fulvestrant and palbociclib suppress ER and pRB but only the triple blocks ERK1/2, ER and pRB. However, it is interesting that palbociclib combined with fulvestrant appears to lead to re-expression of ER compared to fulvestrant alone. Do the authors have any comment regarding this and similarly suppression of FGFR1 also appears to lead to increased ER expression.

The triple drug combinations are then explored in a xenograft model, which requires some clarification. As I understand, mice were implanted with short-term E2 pellets to support proliferation of the MCF7eGFP and MCF7FGFR1 cell lines. Pellets were then removed providing an AI-background yet the MCF7eGFP continue to proliferate; are they E-independent? Furthermore, the clinical background modelled is AI, plus fulvestrant, combined with CDK4/6 inhibition and or FGFR1 inhibition?

The comparisons seem selective in the reporting. Comparison of the xenograft suppl fig 4A and Fig 3A do indeed show that the triple combination is superior, however, lucitinib plus fulvestrant appears more effective than palbociclib plus fulvestrant and single agent lucitinib appears more effective than fulvestrant. This does not detract from the efficacy of the triple combination and enhances the authors argument. Perhaps a small table of comparisons could be added.

The authors then explore the triple combination in three further FGFR amplified cell lines CAMA1, MDA-MB-134 and HC1500 with a particular focus on CAMA1. Here they shown siFGFR in combination with fulvestrant and palbociclib can phenocopy the antiproliferative of the triple drug combination. However, the cell cycle analysis is a little confusing as palbo plus fulvestrant appears to more cells in S-phase than either vehicle or fulvestrant. Additionally the authors suggest (suppl 7) that lucitanib alone or combined with fulvestrant has no impact on cell cycle and yet lucitanib monotherapy appears to decrease the number of cells in G0/G1 and increase those in S-phase?

The authors then address alterations in gene expression in response to palbociclib plus fulvestrant versus the triple combination and provide evidence for an FGFR-RAS/RAF/MEK/ERK signalling access and unsurprisingly the triple drug combination decreases proliferation signatures (CCND1), estrogen early response and mesenchymal pathways.

Based on these observations the authors go on to KD CCND1 to phenocopy the antiproliferative activity of the triple combination. From this they draw the conclusion that FGFR1-CCND1 axis is responsible for poor response to CDK4/6 and fulvestrant.

Based on this the authors test the triple combination in an FGFR1/CCND1 amplified PDX model. Here an alternate FGFR inhibitor is used erdafitinib. The authors state the effect on CAMA1 is similar to lucitanib which again is incorrect as erdafitinib shows a significant antiproliferative effect as a monotherapy and is a pan FGFR inhibitor. The authors state they switched agents as lucitanib targets VEGFR yet it was used in the initial xenograft studies?

Nanostring analysis of tumours resected at the end of study using the panCancer panel showed fulvestrant plus palbociclib increased expression of FGFR1, FGF12 and FGF13. Unfortunately the authors do not have material for the triple combination at end of study however, they do have samples from the PD analysis and it would be interesting to determine if this is a rapid rewiring in response to therapy and if the triple combination can suppress this.

Finally the authors carry out a translational analysis on a small cohort of patients and while interesting, as they rightly point out themselves much larger numbers are required to conclude a role for FGFR amplification/signalling in resistance to CDK4/6 inhibition combined with endocrine therapy. Nonetheless the data does show a trend.

The authors also conclude that multiple publications are highlighting a role of PI3K/AKT pathway in resistance, indeed the current group were one of the first to suggest targeting PI3K. It would be interesting to see, in vitro in the FGFR models if perturbation of PI3K provides greater efficacy than targeting FGFR.

Minor comments

Supplemental Fig S1B need to align titles on WB

Figure 1D states 8 point drug screen but supplemental 1C shows 5 concentrations

Figure 2C blot should be improved as many are over-exposed making interpretation difficult

Figure 2 states the assays for proliferation were in 6 well plates but methods state 48 wells

Response of CAMA1 cells to lucitanib is highly variable throughout the manuscript. Whilst this does not detract from the message of the triple combination, it appears as a single agent lucitanib increases proliferation in Fig4B, increase in Fig 8B and decrease in Suppl Fig 6. If these are multiple biological replicates there should be some consistency

Supplemental figure 8 A colony assay is overgrown making interpretation difficult

Reviewer #2 (Remarks to the Author):

This manuscript describes that FGFR1 amplification is a resistance mechanism to CDK4/6 inhibitor and endocrine therapy and that combined treatment with FGFR, CDK4/6, and antiestrogens is a feasible therapeutic strategy in these tumors. The authors performed a gain-of-function ORF screen for kinases in MCF7 cells treated with CDK4/6 inhibitor and fulvestrant and identified FGFR1 among other kinases. Using various assays they show that FGFR activation upregulates cyclin D1 and overcomes CDK4/6 inhibitor-induced arrest. They also show that in patients ctDNA FGFR amplification is associated with shorter progression-free survival.

Specific comments:

1. FGFR1 amplification was previously described as resistance mechanism to endocrine therapy (Turner et al. Cancer Res. 2010). The authors themselves identified FGFR1 as a hit in the ORF screen fulvestrant -only arm. In most experiments the authors do not test double combinations for each of the 3 drugs and also the palbo-only arm is missing in some cases. It would be important to show all dual and single agents in all key experiments (like xenografts) and also do real synergy/addictiveness testing of all drugs to ensure that the FGFR1 gain is not only synergistic with fulvestrant or palbo alone.
2. The authors use multiple different CDK4/6 and FGFR inhibitors in the various experiments with different cell lines. Since each compound is somewhat different (specificity and off target effects), it would be important to complete all key experiments with the same compounds.
3. The authors show that 41% of patients who progress on fulvestrant +palbo therapy have FGFR amplification based on ctDNA of serum. It is not so easy to detect copy number gain in ctDNA, what controls did the author use? Do these patient primary tumor and metastases also have FGFR amplification at least in a subset of cells? They could do FISH on FFPE slides.

Reviewers' comments:

Reviewer #1 (Remarks to the Author):

In the following manuscript the authors expressed the ORFs of 559 kinases in MCF7 cells treated with fulvestrant ± ribociclib and identified 11 kinases which increased cell viability by greater than 30%. Five kinases (FGFR1, FRK, HCK, FGR, CRKL) also induced resistance in secondary screens. Transduction of MCF7 and T47D with FGFR1 reduced response to ribociclib ± fulvestrant, an observation also noted in FGFR1-amplified breast cancer cell lines. The authors show treatment of an FGFR1-amplified PDX with fulvestrant, palbociclib or both delayed tumour growth but the addition of erdafitinib as a triple combination resulted in complete tumour response. The authors provide correlative evidence that treatment of FGFR1-amplified cells with FGF2, enhances expression of CCND1 associated gene signature and that siRNA of CCND1 increased sensitivity of FGFR1-amplified cells to fulvestrant and palbociclib. The authors proceed to show using NGS of ctDNA from 34 patients pre and post progression on CDK4/6 inhibitors that activating mutations in FGFR1/2 were evident in 41% of the post-progression samples. They then show that ctDNA analysis of MONALEESA-2 showed patients with FGFR1 amplification show PFS of 10.61 months versus 24.84 months in those harbouring wt-FGFR. The authors conclude that ER+ breast cancers, which harbour FGFR1 amplification are resistant to fulvestrant and CDK4/6 inhibitors and may benefit from a triple combination targeting ER, CDK4/6 and FGFR1 signalling.

Major Comments

- 1. The manuscript is well written but is largely correlative and the authors themselves highlight many limitations within their study particularly with regard to the translational science using MONALEESA-2.***

We agree with the Reviewer that this body of work does not prove a causal association between FGFR pathway alterations and drug resistance. Moreover, to demonstrate the translational relevance of this casual association, it would be necessary a clinical trial with a FGFR inhibitor where appropriately selected patients, based on presence of FGFR alterations in their cancers, are randomized to standard of care antiestrogen plus a CDK4/6 inhibitor ± a specific FGFR inhibitor. We will be initiating such randomized trial but, first, we are in the middle of a phase Ib trial of the combination of fulvestrant, palbociclib and the FGFR1-4 inhibitor erdafitinib (NCT03238196) to assess safety and pharmacodynamics inhibition of FGFR signaling.

- 2. With regard to target identification using the ORF 559 kinase library it is of interest that FGR, HCK, FRK all members of the src family, which when over expressed showed decreased sensitivity to ribociclib and palbociclib yet they were omitted from further study by nature of the fact dasatinib has not shown utility in endocrine resistant breast cancer patients and therefore the authors selected FGFR1 as their preferred target which they support by using TCGA and the fact that FGFR1 amplification is already associated with resistance to tamoxifen. The authors go on to test the antiproliferative effect of lucitanib an FGFR1 and VEGFR inhibitor. They show that a triple combination of CDK4/6 inhibition with fulvestrant shows greatest effect (Fig 2). However, the***

statement that FGFR1 inhibition alone is ineffective in the colony forming assays shown in Suppl 2 is a little misleading as it appears there is a least a 40% reduction in the MCF7^{FGFR1}, yet the T47D^{FGFR1} is quantified and the WB confirming expression is also only shown for T47D. The data should be provided for both overexpressing cell lines. Nonetheless the triple does indeed appear more effective. The authors then state that fulvestrant and palbociclib suppress ER and pRb but only the triple blocks ERK1/2, ER and pRb. However, it is interesting that palbociclib combined with fulvestrant appears to lead to re-expression of ER compared to fulvestrant alone. Do the authors have any comment regarding this and similarly suppression of FGFR1 also appears to lead to increased ER expression.

We thank the Reviewer for pointing this out. For this revised manuscript, we have enhanced the Figure 2 (with FGFR1 levels in T47D^{FGFR1} cells) and deleted Suppl. Fig. 2. We have repeated all the immunoblot in the figure, and we didn't see the modest re-expression of ER after treatment with fulvestrant or FGFR-inhibitors.

Figure 2: FGFR1 overexpression confers resistance to fulvestrant and palbociclib. **A.** Tile plot of ER+ breast cancers in TCGA (Cell 2015) with amplification and/or mRNA upregulation of FGFR1, CRKL, HCK, FRK and FGR. **B-D.** MCF-7^{eGFP} and MCF-7^{FGFR1} cells were seeded in 6-well plates in full media supplemented with 2 ng/mL FGF2 and treated with vehicle (DMSO), 1 μ M fulvestrant, or 1 μ M fulvestrant plus 1 μ M palbociclib \pm 1 μ M lucitanib. Drugs and media were replenished every 3 days. After 14 days, monolayers were stained with crystal violet and analyzed as described in Methods. Quantification of the integrated intensity values as fold change relative to vehicle-treated controls are shown (**C**) (**p<0.01 vs. controls, Student's t-test). Cell lysates were subjected to immunoblot analyses with the indicated antibodies (**D**). **E.** T47D^{eGFP} and T47D^{FGFR1} cells were seeded in 6-well plates in full media supplemented with 2 ng/mL FGF2 and treated with vehicle (DMSO), 0.5 μ M fulvestrant, or 0.5 μ M fulvestrant plus 0.5 μ M palbociclib \pm 1 μ M lucitanib. Drugs and media were replenished every 3 days. After 14 days, plates were washed and stained with crystal violet; imaging intensity was quantified by spectrophotometric detection. Quantification of the integrated intensity values as fold change relative to vehicle-treated controls are shown (**p<0.01 vs. controls, Student's t-test). Lysates from T47D cells stably transduced with a FGFR1 expression vector were subjected to immunoblot analysis with FGFR1 and actin antibodies (on top-right of the panel).

3. The triple drug combinations are then explored in a xenograft model, which requires some clarification. As I understand, mice were implanted with short-term E2 pellets to support proliferation of the MCF7eGFP and MCF7FGFR1 cell lines. Pellets were then removed providing an AI-background yet the MCF7eGFP continue to proliferate; are they E-independent? Furthermore, the clinical background modelled is AI, plus fulvestrant, combined with CDK4/6 inhibition and or FGFR1 inhibition? The comparisons seem selective in the reporting. Comparison of the xenograft suppl fig 4A and Fig 3A do indeed show that the triple combination is superior, however, lucitanib plus fulvestrant appears more effective than palbociclib plus fulvestrant and single agent lucitanib appears more effective than fulvestrant. This does not detract from the efficacy of the triple combination and enhances the authors argument. Perhaps a small table of comparisons could be added.

Thanks to the reviewer for pointing this out. We use the 14 days E2 pellet to support the initial growth of ER+ MCF-7 xenografts. However, MCF-7 cells also have other alterations such as PIK3CA activating mutation. We speculate that, once the tumors get established, oncogenic co-mutations such as PIK3CA can sustain the progression of ER+ tumors even in the presence of low levels of circulating estrogens. Moreover, the use of fulvestrant has the intent to suppress the function of the estrogen receptor entirely. This technique is well established and routinely used in our laboratory and others:

- Kinome-Wide RNA Interference Screen Reveals a Role for PDK1 in Acquired Resistance to CDK4/6 Inhibition in ER-Positive Breast Cancer. Jansen VM et al. Cancer Research 2017
- ER α -dependent E2F transcription can mediate resistance to estrogen deprivation in human breast cancer. Miller TW Cancer Discovery 2011

We have added, as the reviewer suggested, a small table of comparisons in Suppl. Fig. 2A

4. **The authors then explore the triple combination in three further FGFR amplified cell lines CAMA1, MDA-MB-134 and HC1500 with a particular focus on CAMA1. Here they shown siFGFR in combination with fulvestrant and palbociclib can phenocopy the antiproliferative of the triple drug combination. However, the cell cycle analysis is a little confusing as palbo plus fulvestrant appears to more cells in S-phase than either vehicle or fulvestrant. Additionally, the authors suggest (suppl 7) that lucitanib alone or combined with fulvestrant has no impact on cell cycle and yet lucitanib monotherapy appears to decrease the number of cells in G0/G1 and increase those in S-phase?**

We have repeated the experiment after 12 h of treatment and confirmed that the triple combination is the best in blocking the cells in G0/G1 (94% vs 84% of fulvestrant/palbociclib). Moreover, we confirmed that treatment with lucitanib increases the cells in phase S of the cell cycle. We cannot explain that at this time but this is in line with the inefficacy of lucitanib as monotherapy. We have now changed Fig. 4D as follows:

5. **The authors then address alterations in gene expression in response to palbociclib plus fulvestrant versus the triple combination and provide evidence for a FGFR-RAS/RAF/MEK/ERK signaling access and unsurprisingly the triple drug combination decreases proliferation signatures (CCND1), estrogen early response and mesenchymal pathways. Based on these observations the authors go on to KD CCND1 to phenocopy the antiproliferative activity of the triple combination. From this they draw the conclusion that FGFR1-CCND1 axis is responsible for poor response to CDK4/6 and fulvestrant. Based on this the authors test the triple combination in an FGFR1/CCND1 amplified PDX model. Here an alternate FGFR inhibitor is used erdafitinib. The authors state the effect on CAMA1 is similar to lucitanib which again is incorrect as erdafitinib shows a significant antiproliferative effect as a monotherapy and is a pan FGFR inhibitor. The authors state they switched agents as lucitanib targets VEGFR yet it was used in the initial xenograft studies?**

We thank the reviewer for this annotation. Erdafitinib is a pan-FGFR inhibitor with more potency than lucitanib as shown in the table below. The original studies were done with lucitanib, before erdafitinib was available to us. Precisely to address our concern that lucitanib worked at least in part via VEGFR *in vivo*, we switched

the 2nd set of studies to the more specific FGFR inhibitor. Obviously, this was less of a concern for the studies with cells in culture.

	Erdafitinib IC ₅₀ (nM)	Lucitanib IC ₅₀ (nM)
FGFR1	< 1	21
FGFR2	< 1	41
FGFR3	1.05	51
FGFR4	< 1	-----
FGFR3 (G697C)	1.90	-----
VEGFR1	-----	1
VEGFR2	-----	1.1
VEGFR3	-----	7.1
PDGFR α	-----	0.43
PDGFR β	-----	0.26

We performed again the clonogenic assay with erdafitinib at 10 nM. As shown in new Suppl. Fig. 9A, the effects on CAMA1 of erdafitinib and lucitanib is similar.

Supplementary Figure 9. A. CAMA1 cells plated in full media containing FGF2 were treated with vehicle, 1 μ M fulvestrant, 1 μ M palbociclib, 0.01 μ M erdafitinib, each alone or in combination as indicated. Drugs and media were replenished every 3 days. Fourteen days later, monolayers were stained with crystal violet and analyzed as described in Methods. Quantification of the intensity values as fold change relative to vehicle-treated controls are shown (**** p <0.0001 vs. controls, Student's t-test). **B.** CAMA1 cells were treated with vehicle, 1 μ M fulvestrant, 1 μ M palbociclib, 0.25 μ M erdafitinib for 6 h. Cell lysates were prepared and subjected to immunoblot analysis with the indicated antibodies.

6. **Nanostring analysis of tumours resected at the end of study using the panCancer panel showed fulvestrant plus palbociclib increased expression of FGFR1, FGF12 and FGF13. Unfortunately, the authors do not have material for the triple combination at end of study however, they do have samples from the PD analysis and it would be interesting to determine if this is a rapid rewiring in response to therapy and if the triple combination can suppress this.**

We thank the reviewer for pointing this out. We were indeed able to collect 3 tumors per group from mice treated for only one week with vehicle, fulvestrant plus palbociclib, and the triple combination. One tumor in each group was exhausted in other assays but we were able to extract RNA in 2/3 tumors from each group. With only 2 tumors, it is hard to draw statistical conclusions but we would like to share these data with the reviewer. Our results confirm that after one week of treatment there was an increase of expression of FGF12/13 and FGFR1 mRNA in tumors treated with the doublet of fulvestrant and palbociclib. This increase was more evident in tumors treated with the triple combination where we saw an increase in expression of FGFR1, FGFR2, and all 22 FGFs ligands, also suggesting a potential compensatory effect to the inhibition of FGFR function. Conversely, CCND1 expression was reduced only in the group treated with the triple combination. We include these data below for consideration of the Reviewer. Will be happy to include these as supplementary data and comment on them upon editorial advice.

7. ***Finally, the authors carry out a translational analysis on a small cohort of patients and while interesting, as they rightly point out themselves much larger numbers are required to conclude a role for FGFR amplification/signalling in resistance to CDK4/6 inhibition combined with endocrine therapy. Nonetheless the data does show a trend. The authors also conclude that multiple publications are highlighting a role of PI3K/AKT pathway in resistance, indeed the current group were one of the first to suggest targeting PI3K. It would be interesting to see, in vitro in the FGFR models if perturbation of PI3K provides greater efficacy than targeting FGFR.***

We would like to share with the reviewer unpublished data from our group suggesting perturbation of PI3K is not as effective as inhibition of FGFR1. CAMA-1 cells in 10% DMEM-FBS were treated with the PI3K α inhibitor alpelisib, the MEK inhibitor selumetinib and lucitanib, each alone and in combination with fulvestrant, and paclitaxel alone. As shown in panels A, B below, alpelisib \pm fulvestrant was not

effective. Moreover, we previously reported FGFR1 as possible mechanisms of resistance to the PI3K inhibitors (A Phase Ib Study of Alpelisib (BYL719), a PI3K α -Specific Inhibitor, with Letrozole in ER+/HER2- Metastatic Breast Cancer., Mayer et al Clinical Cancer Research 2017).

Minor comments

Supplemental Fig S1B need to align titles on WB. Figure 1D states 8-point drug screen but supplemental 1C shows 5 concentrations. Figure 2 states the assays for proliferation were in 6 well plates but methods state 48 wells. Figure 2C blot should be improved as many are over-exposed making interpretation difficult. Supplemental figure 8 A colony assay is overgrown making interpretation difficult.

We thank the Reviewer for pointing this out. We have now corrected all these in the revised manuscript. We hope new panels are satisfactory.

Response of CAMA1 cells to lucitanib is highly variable throughout the manuscript. Whilst this does not detract from the message of the triple combination, it appears as a single agent lucitanib increases proliferation in Fig4B, increase in Fig 8B and decrease in Suppl Fig 6. If these are multiple biological replicates there should be some consistency.

We appreciated the feedback from the reviewer and we have repeated the growth assay in supplementary figure 5 (suppl fig 6 in the old version of the manuscript) and corrected the figure as showed below:

Figure 4B

**Supl. Figure 6B
(Suppl. Fig. 8B old version)**

**Supl. Figure 5
(Suppl. Fig. 6 old version)**

Reviewer #2 (Remarks to the Author):

This manuscript describes that FGFR1 amplification is a resistance mechanism to CDK4/6 inhibitor and endocrine therapy and that combined treatment with FGFR, CDK4/6, and antiestrogens is a feasible therapeutic strategy in these tumors. The authors performed a gain-of-function ORF screen for kinases in MCF7 cells treated with CDK4/6 inhibitor and fulvestrant and identified FGFR1 among other kinases. Using various assays they show that FGFR activation upregulates cyclin D1 and overcomes CDK4/6 inhibitor-induced arrest. They also show that in patients ctDNA FGFR amplification is associated with shorter progression-free survival.

Specific comments:

- 1. FGFR1 amplification was previously described as resistance mechanism to endocrine therapy (Turner et al. Cancer Res. 2010). The authors themselves identified FGFR1 as a hit in the ORF screen fulvestrant-only arm. In most experiments the authors do not test double combinations for each of the 3 drugs and also the palbo-only arm is missing in some cases. It would be important to show all dual and single agents in all key experiments (like xenografts) and also do real synergy/addictiveness testing of all drugs to ensure that the FGFR1 gain is not only synergistic with fulvestrant or palbo alone.***

We thank the reviewer for pointing this out. We agree with the Reviewer on the well-described role of FGFR1 in antiestrogen resistance in early ER+ breast cancer as published by us and others [Turner N et al Cancer Res. 2010 Mar 1;70(5):2085-94; Formisano L et al in Clinical Cancer Res. 2017 Oct 15;23(20):6138-50; Giltnane JM et al in Sci. Transl. Med. 2017 Aug 9;9(402)]. We note, however, these studies were conducted in patients with early but not advanced ER+ breast cancer, where CDK4/6 inhibitors are now approved. Of course the impact per se of aberrant FGFR signaling on antiestrogen action is part of the effect we are seeing but, the point we are trying to make is that this effect is maintained for the combination of antiestrogens and CDK4/6 inhibitors.

We did not combine the FGFR inhibitor with palbociclib alone mainly because dispensing with the antiestrogen first may be challenging as a clinical strategy at least for now. Indeed, we are in the middle of a phase Ib trial of the combination of fulvestrant, palbociclib and erdafitinib (NCT03238196) to assess safety and pharmacodynamics inhibition of FGFR signaling. To satisfy the reviewer request, we performed a clonogenic assay with all single drug and their combinations. We confirmed the triple combination with fulvestrant, palbociclib and lucitanib is the more effective to inhibit CAMA1 cell growth.

2. The authors use multiple different CDK4/6 and FGFR inhibitors in the various experiments with different cell lines. Since each compound is somewhat different (specificity and off target effects), it would be important to complete all key experiments with the same compounds.

We used multiple inhibitors – all FDA approved – precisely to avoid the limitation of testing only one. Further, similar results with all three suggest to us the treatment effects are likely specific to CDK4/6 inhibition. However we cannot rule out subtle differences in specificities without a major amount of significant work. We are aware that palbociclib and ribociclib may have a slightly different pharmacological spectrum (CDK4/6) than abemaciclib (CDK4/6/9; Chen et al. Spectrum and Degree of CDK Drug Interactions Predicts Clinical Performance, Molecular Cancer Therapeutic 2016). We discussed above the reasons to add erdafitinib as a second inhibitor, that is to address our concern that lucitanib worked at least in part via VEGFR *in vivo*.

3. The authors show that 41% of patients who progress on fulvestrant +palbo therapy have FGFR amplification based on ctDNA of serum. It is not so easy to detect copy number gain in ctDNA, what controls did the author use? Do these patient primary tumor and metastases also have FGFR amplification at least in a subset of cells? They could do FISH on FFPE slides.

We sincerely thank the Reviewer for these comments. We recognize the limits to detect gene amplification on ctDNA. In our assay we used previously published internal controls (Odegaard JJ, Vincent JJ, Mortimer S, Vowles JV, Ulrich BC, Banks KC, et al. Validation of a plasma-based comprehensive cancer genotyping assay utilizing orthogonal tissue- and plasma-based methodologies. Clin. Cancer Res. 2018) also described in the Methods section Six of 9 tumors from patients with FGFR1 amplification in ctDNA were available for FISH (FGFR1:CEN8 ratio) or targeted capture next generation sequencing (NES). All of 6 patients showed FGFR1 amplification. We have added these data as text in Results.

We sincerely hope that with these responses, new data and modifications, we have been responsive to the Reviewers. Again, we appreciate their input and suggestions, which have substantially improved our manuscript. We respectfully request a second review and consideration for publication in *Nature Communications*. Thanks for your time and consideration.

Sincerely,

Carlos L. Arteaga

REVIEWERS' COMMENTS:

Reviewer #1 (Remarks to the Author):

The following manuscript describes the role of FGFR amplification in resistance to CDK4/6 inhibition. The authors have answered my major queries and supplied supporting data and I am happy with their responses.

Reviewer #2 (Remarks to the Author):

The authors have responded to each of the reviewers' major points and revised the manuscript accordingly. The inclusion of the additional data and the revisions significantly strengthened the manuscript.

December 05, 2018

RE: Resubmission of Manuscript NCOMMS-18-20128 '*Aberrant FGFR pathway signaling mediates resistance to CDK4/6 inhibitors in ER+ breast cancer*'.

Reviewer #1 (Remarks to the Author):

The following manuscript describes the role of FGFR amplification in resistance to CDK4/6 inhibition. The authors have answered my major queries and supplied supporting data and I am happy with their responses.

Reviewer #2 (Remarks to the Author):

The authors have responded to each of the reviewers' major points and revised the manuscript accordingly. The inclusion of the additional data and the revisions significantly strengthened the manuscript.

We thank the referees for their input and suggestions, which have substantially improved our manuscript.

Sincerely,

Carlos L. Arteaga